# ReDit: Reward Dithering for Improved LLM Policy Optimization

**Chenxing Wei**[†§♮], **Jiarui Yu**[°], **Ying He**[†], **Hande Dong**[°], **Yao Shu**[ℓ*], **F. Richard Yu**[‡]

[§]Guangdong Lab of AI and Digital Economy (SZ), China
[†]College of Computer Science and Software Engineering, Shenzhen University, China
[ℓ]Hong Kong University of Science and Technology (Guangzhou), China
[°] Tencent, Shenzhen, China
[‡]School of Information Technology, Carleton University, Canada
`weichenxing2023@email.szu.edu.cn, yaoshu@hkust-gz.edu.cn`

## Abstract

DeepSeek-R1 has successfully enhanced Large Language Models (LLMs) reasoning capabilities through its rule-based reward system. While it's a "perfect" reward system that effectively mitigates reward hacking, such reward functions are often discrete. Our experimental observations suggest that discrete rewards can lead to gradient anomaly, unstable optimization, and slow convergence. To address this issue, we propose ReDit (Reward Dithering), a method that dithers the discrete reward signal by adding simple random noise. With this perturbed reward, exploratory gradients are continuously provided throughout the learning process, enabling smoother gradient updates and accelerating convergence. The injected noise also introduces stochasticity into flat reward regions, encouraging the model to explore novel policies and escape local optima. Experiments across diverse tasks and different LLMs demonstrate the effectiveness and efficiency of ReDit. On average, ReDit achieves performance comparable to vanilla GRPO with only approximately 10% the training steps, and furthermore, still exhibits a 4% performance improvement over vanilla GRPO when trained for a similar duration. Visualizations confirm significant mitigation of gradient issues with ReDit. Moreover, theoretical analyses are provided to further validate these advantages.

## 1 Introduction

Reinforcement learning (RL) is pivotal in Large Language Models (LLMs) development [1, 2, 3, 4]. Initially, RL from human feedback (RLHF) [5, 6] was employed to align pre-trained LLMs with human preferences [7, 8]. This typically involves training a separate reward model (RM) on human preference data [9], which then guides the LLM policy optimization [10]. While effective, this approach introduces considerable training overhead [11]. Subsequently, methods like Direct Preference Optimization (DPO) [12, 13] were developed, enabling LLMs to learn directly from preference data and thus bypassing explicit RM training. However, these methods still require extensive collection of high-quality preference data. For reasoning tasks such as mathematics and coding, DeepSeek-R1 [14] with Group Relative Policy Optimization [15](GRPO) proposes an alternative: optimizing the LLM policy directly using a rule-based reward system [16, 17], thereby avoiding the need for external RMs or large preference datasets. For instance, such a system might assign a reward of 1 for outputs meeting predefined criteria (e.g., correctness, format compliance) and 0 otherwise [14]. The simplicity and unbiased nature of these rule-based rewards prevent LLMs from hacking them, potentially fostering enhanced reasoning capabilities [18].

---

[*]corresponding author. [♮] Work done during an internship at Tencent.

39th Conference on Neural Information Processing Systems (NeurIPS 2025).

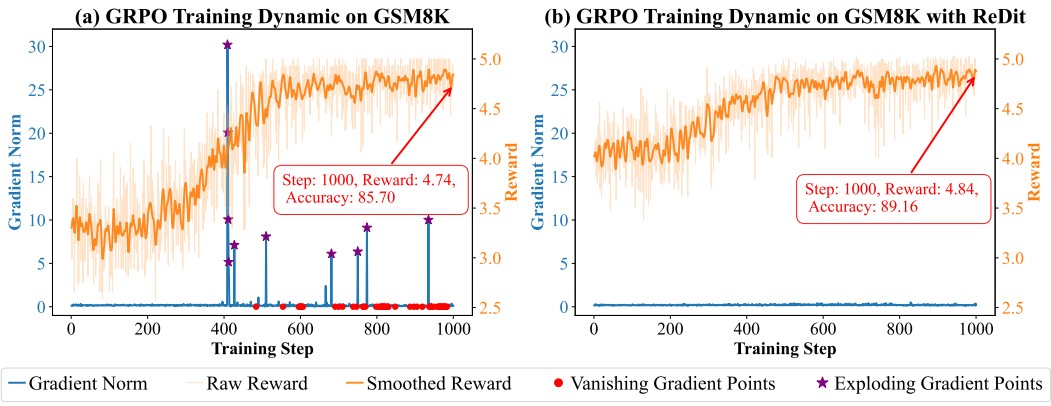

Figure 1: Training Dynamics of Gradient Norm and Reward for Qwen2.5-7B [31] on GSM8K [32] Dataset. The left and right figures compare original gradient norm (before gradient clipping [33]) and reward trends across training steps. The original GRPO method (the left figure) suffers from significant gradient instability—both vanishing (red dots, norms < 0.01) and exploding (purple asterisks, norms > 5). In contrast, ReDit with Gaussian reward smoothing (the right figure) effectively stabilizes optimization throughout training.

However, such reward functions are often discrete, posing significant optimization challenges [19, 20, 21]. Consider an RL scenario with a binary reward [22]: a policy model receives 1 for a correct answer and 0 otherwise. During early training phases, a policy LLM rarely generates completely correct answers, resulting in predominantly zero rewards across mini-batches [23]. Although the model may engage in exploratory behavior on difficult examples, the corresponding gradients remain minimal due to small advantage magnitudes [24]. Thus, these hard examples and potentially beneficial explorations [24] are largely unexploited during the early stages. Conversely, the model may repeatedly reinforce easy examples [25], thus reducing incentives to explore alternative strategies for more difficult problems [26]. This phenomenon can lead to training stagnation in intermediate and advanced stages. Consistent with this, as shown in Fig. 1 (the left figure), we observe that the policy model frequently suffers from gradient vanishing [27, 28] or explosion [29, 30] during these phases. This combination of insufficient exploration and gradient instability substantially impedes model convergence, representing a critical obstacle to efficient RL in LLMs.

This observed phenomenon highlights that even perfectly accurate discrete reward functions face significant limitations within gradient-based optimization frameworks. Lending theoretical support to this, recent studies [34, 35, 36] have established that a singular focus on increasing reward model accuracy does not necessarily translate to enhanced language model performance. In particular, [36] theoretically substantiates the necessity for effective reward models to integrate adequate variance and uncertainty to enable efficient optimization. The theoretical details are given in Sec. 3.2. Consequently, we believe that an excellent reward system should achieve a careful equilibrium between correctness and sufficient variance.

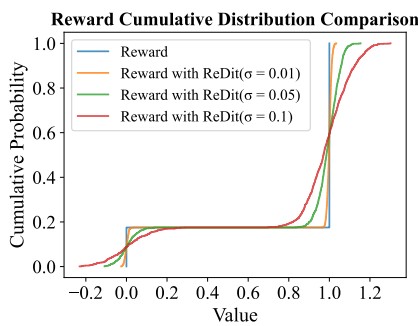

Figure 2: The figure illustrates how ReDit of different variances gradually smooth the reward distribution, showing the smoothing effect of perturbations of different variances on the reward distribution.

Inspired by these observations and theoretical insights, we propose ReDit, a simple but effective technique that introduces zero-mean random perturbations to discrete reward signals during training. By adding small random noise to the reward function (Fig. 2), ReDit effectively softens the original hard reward function. Compared to a hard reward function, a softened one can provide greater reward variance within mini-batches,

which, as indicated by previous research, can lead to enhanced model performance and accelerated convergence.

Fig. 1 (the right figure) illustrates the impact of our proposed ReDit on LLM policy optimization for the GSM8K [32] task. The orange lines indicate that during early training phases, GRPO with ReDit achieves significantly higher average rewards compared to the baseline (GRPO without ReDit), demonstrating the efficacy of our method. We hypothesize that ReDit encourages broader exploration by assigning varied rewards to outputs that only partially meet strict evaluation criteria, thereby accelerating convergence. Towards the end of training (1000 steps), while both policy models attain high rewards, our approach with ReDit exhibits superior performance on the test set, indicating enhanced generalization. Additionally, ReDit demonstrates more robust gradient updating. As shown in Fig. 1 (the left figure), phenomena such as gradient vanishing (red point) and explosion (purple star) emerge during training with the baseline. In contrast, GRPO with ReDit maintains stable gradients throughout the training process. These findings highlight the advantages of ReDit: more stable policy optimization, faster convergence, and improved overall performance.

Moreover, theoretical analysis indicates that a greater reward variance can enhance performance and accelerate convergence in reinforcement learning [36]. We increase reward variance within mini-batches while preserving the expected gradient through reward dithering. By carefully injecting noise into the reward function, ReDit achieves a balance between reward signal fidelity and reward variance, leading to enhanced policy optimization.

In summary, our main contributions are:

- We observe that policy optimization under discrete reward functions suffer from unstable gradients and slow convergence(Section 3.1).
- We propose Reward Dithering (ReDit), a simple yet effective technique that introduces perturbations to discrete rewards. This method is shown to accelerate convergence speed and enhance final model performance (Algorithm 1 and Section 4).
- Extensive experiments across diverse downstream tasks, reinforcement learning algorithms, and perturbation distributions demonstrate that ReDit achieves superior performance and enhanced convergence (Section 5).
- Theoretical analysis proves that ReDit produces an unbiased estimate of the original gradient (Proposition 1) and introduces beneficial gradient variance that mitigates vanishing and exploding gradients (Proposition 2).

## 2   Preliminaries

We frame LLM generation as a sequential decision-making problem solvable via RL. The process is modeled as a Markov Decision Process (MDP) [37] where the state $s_t = q; o_{<t}$ includes the prompt $q$ and generated tokens $o_{<t}$, the action $o_t$ is the next token selected from the vocabulary, and the policy $\pi_\theta(o_t|s_t)$ is parameterized by $\theta$. The goal is to optimize the policy to maximize the expected sequence-level reward $R(q, o) = \sum_{t=1}^{|o|} r(s_t, o_t)$ over the prompt distribution $p_Q$:

$$J(\pi_\theta) = \mathbb{E}_{q \sim p_Q} \left[ \mathbb{E}_{o \sim \pi_\theta(\cdot|q)}[R(q, o)] \right]. \tag{1}$$

Recently, GRPO [15] was proposed as a PPO alternative that eliminates the need for independent RMs and value functions. GRPO typically processes sparse, discrete rewards directly, rather than continuous RM scores. For tasks like mathematical reasoning, this discrete reward $R(q, o) \in \{0, 1\}$ is often determined by a simple function checking correctness or format. GRPO estimates the advantage $\hat{A}_{i,t}^{\text{GRPO}}$ by sampling $G$ responses $\{o_i\}_{i=1}^{G}$ and normalizing their discrete rewards within the set. Its objective function, which includes a KL divergence term $D_{\text{KL}}(\pi_\theta||\pi_{\text{ref}})$ for stability, is given by:

$$J_{\text{GRPO}}(\theta) = \mathbb{E}_{q \sim p_Q} \left[ \frac{1}{G} \sum_{i=1}^{G} \sum_{t=1}^{|o_i|} \min \left( r_{i,t}(\theta)\hat{A}_{i,t}^{\text{GRPO}}, \text{clip}\left(r_{i,t}(\theta), 1 - \epsilon, 1 + \epsilon\right) \hat{A}_{i,t}^{\text{GRPO}} \right) \right]$$
$$- \beta \mathbb{E}_{q \sim p_Q} \left[ D_{\text{KL}}(\pi_\theta(\cdot|q)||\pi_{\text{ref}}(\cdot|q)) \right], \tag{2}$$

where $r_{i,t}(\theta) = \frac{\pi_\theta(o_{i,t}|s_{i,t})}{\pi_{\theta_{\text{old}}}(o_{i,t}|s_{i,t})}$. Subsequent methods such as DAPO [38], Dr.GRPO [39], and REIN-FORCE++ [40] generally adopt this discrete reward paradigm (see Appendix A for more related

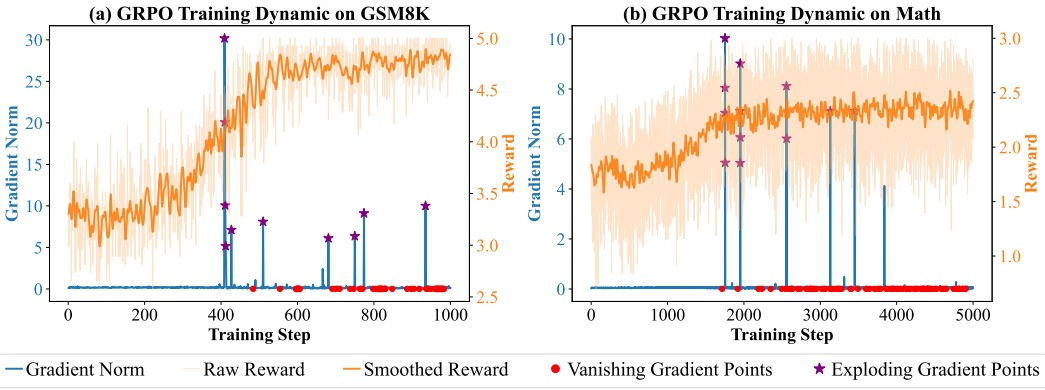

Figure 3: Qwen2.5-7B [31] Gradient norm and reward training dynamics of standard GRPO on GSM8k and MATH datasets. During the whole optimization process, the gradient of standard GRPO is unstable, and there are a lot of gradient vanishing or gradient exploding cases.

work). While simplifying the overall RL process by avoiding complex RMs, this shift to discrete, sequence-level rewards introduces significant optimization challenges. The inherent sparsity and abrupt value changes (e.g., 0 to 1) hinder policy gradient estimation and lead to training instability (Section 3.1).

# 3 Motivation

This section articulates the fundamental motivations driving our research and establishes the critical challenges that our work aims to address. In Section 3.1, we examine the optimization challenges inherent in discrete reward structures, followed by an exposition of the theoretical principles informing our methodological framework in Section 3.2.

## 3.1 Difficulties in Optimization Caused by Discrete Rewards

Optimizing LLM policies using algorithms like GRPO in conjunction with discrete sequence-level rewards (e.g., binary correctness metrics) presents significant optimization challenges. Fig. 3 plots the policy gradient norm (blue line) and average reward (orange line) during standard GRPO training on the GSM8K and MATH datasets, respectively. Two main issues are immediately apparent:

**Gradient Vanishing.** The figure illustrates instances where the gradient norm approaches zero (red dot), occurring when most examples in a GRPO batch yield identical binary rewards. Consequently, the population relative advantage estimate $\hat{A}_{i,t}^{\mathrm{GRPO}}$ becomes negligible across examples, providing insufficient learning signals and causing training stagnation. This phenomenon is evident in Fig. 3(the right figure) post-step 2000.

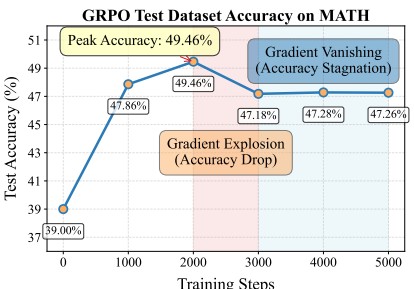

**Gradient Explosion.** Conversely, training dynamics exhibit sporadic sharp spikes in gradient norm (purple asterisks) when small policy changes cause sequences to transition from incorrect (reward 0) to correct (reward 1). These transitions create disproportionately large advantage estimates for newly successful sequences, triggering sudden, destabilizing gradient updates as shown in Fig. 3(the left figure). Such spikes induce reward fluctuations in subsequent steps, hindering smooth convergence and learning efficiency.

Figure 4: GRPO has unstable performance on the MATH test set. The figure plots the test accuracy achieved for the checkpoints saved during the training run shown in Fig. 3(the right figure).

The discrete, sparse rewards induce unstable oscillations between vanishing and exploding gradients. Fig. 4 demonstrates that model performance fluctuates correspondingly with these oscillations. This

inherent instability not only compromises optimization efficiency but also serves as a key motivation for our research.

## 3.2 Theoretical Principles to Address the Limitations of Discrete Rewards

To overcome the critical challenges with discrete rewards outlined in Section 3.1, we propose a direct approach to enhance reward signal quality. Our solution derives from key theoretical insights in policy optimization, particularly Theorem 1 and Theorem 2, which reveal fundamental relationships between reward variance, model accuracy, and learning efficiency. See Appendix B.1 for definitions.

> **Theorem 1** (Policy network optimization time lower bound). *From Theorem 1 in [41]. Suppose that we maximize the objective (Eq. (1)), using a general autoregressive policy* $\pi_\theta(\mathbf{y}|\mathbf{x}) = \prod_{l=1}^{\mathbf{y}} \mathrm{softmax}(f_\theta(\mathbf{x}, \mathbf{y}_{<l})_{\mathbf{y}_l})$. *For any* $\gamma > 0$, *prompt* $\mathbf{x} \in \mathcal{X}$, *and reward function* $r$, *the time it takes until* $\mathbb{E}_{\mathbf{y} \sim \pi_{\theta(t)}(\cdot|\mathbf{x})}[r(\mathbf{x}, \mathbf{y})] \geq \mathbb{E}_{\mathbf{y} \sim \pi_{\theta(0)}(\cdot|\mathbf{x})}[r(\mathbf{x}, \mathbf{y})] + \gamma$ *is:*
>
> $$\Omega \left( \mathbb{E}_{\mathbf{x}' \sim S} \left[ \mathrm{var}_{\mathbf{y} \sim \pi_{\theta(0)}(\cdot|\mathbf{x}')}(r(\mathbf{x}', \mathbf{y})) \right]^{-\frac{1}{3}} \right) \tag{3}$$
>
> *The reward variance is:* $\mathrm{var}_{\mathbf{y} \sim \pi_\theta(\cdot|\mathbf{x})}[r(\mathbf{x}, \mathbf{y})] := \mathbb{E}_{\mathbf{y} \sim \pi_\theta(\cdot|\mathbf{x})}[[r(\mathbf{x}, \mathbf{y}) - \mathbb{E}_{\mathbf{y}' \sim \pi_\theta(\cdot|\mathbf{x})}[r(\mathbf{x}, \mathbf{y}')]]^2]$.

Theorem 1 establishes that the time $t_\gamma$ required for policy improvement is inversely proportional to reward variance. When rewards exhibit insufficient variance—failing to adequately differentiate between high-quality and low-quality outputs under policy $\pi_\theta$, convergence slows significantly. This finding suggests that strategically increasing reward variance can accelerate policy convergence.

> **Theorem 2** (Policy network optimization time upper bound). *From Theorem 2 in [41]. Assume* $\pi_\theta$ *is a policy of the form* $\pi_\theta(\mathbf{y}|\mathbf{x}) = \mathrm{softmax}[\theta_{:,\mathbf{x}}]_{\mathbf{y}}$. *Given a prompt* $\mathbf{x} \in S$, *let* $\gamma > 0$ *and denote by* $t_\gamma > 0$ *the initial time of* $\mathbb{E}_{\mathbf{y} \sim \pi_{\theta(t)}(\cdot|\mathbf{x})}[r_{\mathrm{G}}(\mathbf{x}, \mathbf{y})] \geq \mathbb{E}_{\mathbf{y} \sim \pi_{\theta(0)}(\cdot|\mathbf{x})}[r_{\mathrm{G}}(\mathbf{x}, \mathbf{y})] + \gamma$. *For any initial policy* $\pi_{\theta(0)}$, *a perfect RM converges to* $t_\gamma$ *that can be arbitrarily large, while a relatively inaccurate RM has an upper bound of* $\mathcal{O}(\pi_{\theta(0)}(y^\gamma|\mathbf{x})^{-1})$.

Complementarily, Theorem 2 demonstrates that effective reward models must incorporate a calibrated degree of uncertainty. This controlled uncertainty creates essential exploration space during early training stages, preventing premature convergence and facilitating more efficient optimization.

While perfectly accurate reward functions resist reward hacking, they paradoxically impede optimization by producing discrete rewards with minimal variance and insufficient randomness. This limitation severely constrains the growth rates of both training reward $r_{RM}$ and true reward $r_G$ during policy gradient updates. To address this fundamental tension, we introduce ReDit—a method that injects zero-mean perturbations into discrete rewards. This approach preserves the expected reward value while introducing beneficial variance and controlled uncertainty in each update step, dramatically improving both model performance and convergence speed.

## 4 Reward Dithering (ReDit)

As discussed previously, the discrete nature of rewards commonly used in GRPO can lead to unstable gradient dynamics. To address this, we propose **ReDit** . The core idea, detailed in Algorithm 1, is to inject calibrated, zero-mean perturbations into the discrete rewards obtained from sampled outputs before using them to compute the GRPO objective for policy updates. Importantly, our ReDit method preserves the overall optimization structure of the GRPO objective function as defined in Eq. (2), the optimization still aims to maximize this objective.

The crucial modification introduced by ReDit lies in *how* the advantage term $\hat{A}_{i,t}^{\mathrm{GRPO}}$ within Eq. (2) is computed. Instead of directly using the raw discrete rewards $r_i = r(o_i)$ obtained for each sampled output $o_i$ in the batch $\{o_i\}_{i=1}^G$ (line 3 in Algorithm 1), we first compute **smoothed rewards** $\tilde{r}_i$. This is done by adding independently sampled zero-mean perturbation $\epsilon_i$ (e.g., from $\mathcal{N}(0, \sigma^2)$ or $\mathcal{U}[-a, a]$) to each discrete reward (line 4 in Algorithm 1):

$$\tilde{r}_i = r_i + \epsilon_i \tag{4}$$

These smoothed rewards $\{\tilde{r}_k\}_{k=1}^{G}$ are then used as the basis for calculating the advantage. GRPO often computes advantage based on the relative performance within the batch, typically involving normalization. With ReDit, the core component of the advantage calculation, which relies on these rewards, is effectively modified as follows:

$$\hat{A}_{i,t}^{\text{GRPO}} \propto \frac{r_i - \text{mean}(\{r_k\}_{k=1}^{G})}{\text{std}(\{r_k\}_{k=1}^{G})} \quad \xrightarrow{\text{ReDit}} \quad \hat{A}_{i,t}^{\text{Dithering}} \propto \frac{\tilde{r}_i - \text{mean}(\{\tilde{r}_k\}_{k=1}^{G})}{\text{std}(\{\tilde{r}_k\}_{k=1}^{G})} \tag{5}$$

Thus, the relative standing of each output $o_i$ within the batch, which informs its advantage $\hat{A}_{i,t}^{\text{GRPO}}$ used in Eq. (2), is determined by the continuous smoothed reward $\tilde{r}_i$ rather than the discrete $r_i$. This substitution transforms the optimization landscape. By introducing continuous variations via $\tilde{r}_i$, the added noise provides informative, non-zero gradients even when discrete rewards $r_i$ are sparse or identical within a batch, mitigating gradient vanishing. It also dampens the sharp changes in expected advantage resulting from small policy shifts affecting discrete outcomes, thus reducing the likelihood of gradient explosion. This overall smoothing effect facilitates a more stable gradient flow, enabling more robust and efficient optimization of the policy $\pi_\theta$ using the GRPO objective (line 5 in Algorithm 1).

---

**Algorithm 1** ReDit within one optimization step

---

1: **Input:** Base policy $\pi_{\theta_{\text{old}}}$; Discrete reward function $r : \mathcal{O} \to \{0, 1, 2, 3, ...\}$; Prompt $q$; Number of samples $G$. Noise parameters: Gaussian std dev $\sigma > 0$ **or** Uniform radius $a > 0$.
2: **Output:** Updated policy $\pi_\theta$.

3: Sample $G$ outputs $\{o_i\}_{i=1}^{G} \sim \pi_{\theta_{\text{old}}}(\cdot \mid q)$ and compute $r_i \leftarrow r(o_i)$ for $i = 1, \ldots, G$.
4: Sample $\epsilon_i \sim \mathcal{N}(0, \sigma^2)$ **or** $\mathcal{U}[-a, a]$ and compute $\tilde{r}_i \leftarrow r_i + \epsilon_i$ for $i = 1, \ldots, G$.// Generate noise and smooth rewards.
5: Compute $J_{\text{GRPO}}$ using $\{\tilde{r}_i\}_{i=1}^{G}$ and $\theta \leftarrow \text{Optimize}(\theta_{\text{old}}, J_{\text{GRPO}}, \tilde{r}_i)$.// Optimization
6: **return** Updated policy $\pi_\theta$.

---

# 5 Empirical Results

This section presents a thorough evaluation of our ReDit framework, assessing its effectiveness and efficiency. We begin by detailing the datasets and experimental configurations in Section 5.1. Subsequently, Section 5.2 provides a comprehensive analysis of the primary findings. To isolate the contributions of key components, we also conduct ablation studies, the results of which are presented in Section 5.3.

## 5.1 Datasets and Setup

To rigorously evaluate the effectiveness of our proposed ReDit framework, we conducted extensive experiments. The specific experimental settings are detailed below.

**Datasets.** Our dataset selection and setup largely follow the methodology of [15], primarily to assess the mathematical reasoning capabilities of the models. This encompasses mathematical problem-solving datasets such as GSM8K [32] and MATH [42], as well as the multimodal geometric reasoning dataset Geometry3K [43]. Each dataset provides distinct training and test splits, which we utilize accordingly for model training and subsequent evaluation. See the Appendix D.1 for details of the dataset.

Table 1: Comparison of the mean and variance of accuracy for different baselines under 9000 steps on GSM8K.

| Name | DAPO | DR.GRPO | REINFORCE++ |
|---|---|---|---|
| Baseline | 87.52 | 86.13 | 86.25 |
| w/ **ours(Gauss)** | **89.34** ($\pm$ 0.04) | **87.69** ($\pm$ 0.06) | **87.96** ($\pm$ 0.03) |
| w/ **ours(Uniform)** | 88.57 ($\pm$ 0.01) | 87.34 ($\pm$ 0.07) | 87.59 ($\pm$ 0.09) |
| $\Delta$ | +1.82 | +1.56 | +1.71 |

**Reward Functions.** We designed dataset-specific reward functions. For the GSM8K dataset, which involves simpler problem structures, we implemented several reward types: accuracy-based, strict format adherence, sort format adherence, integer value correctness, and inference step adherence.

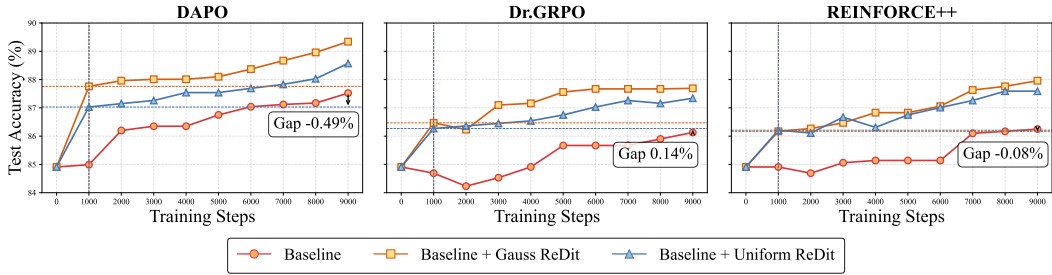

Figure 5: Accuracy of different GRPO variants (DAPO, DR.GRPO, REINFORCE++) tested on the GSM8K dataset. The horizontal dashed line highlights the performance of using ReDit at about 1000 training steps, and even after 9000 steps, its accuracy is comparable to the baseline.

For the more complex MATH and Geometry3K datasets, our supervision relied solely on accuracy-based and inference-based reward functions. Detailed implementations of these reward functions are provided in the Appendix D.2.

**Initial Policy**. To rigorously assess the effectiveness of ReDit without confounding factors introduced by supervised fine-tuning (SFT), we initialized our experiments directly with instruct models without any additional SFT training. Previous research by [44] demonstrated that even random rewards can enhance performance for Qwen models. Therefore, we conducted comprehensive evaluations across a diverse set of instruction-tuned models, including Qwen2.5-7B-Instruct, Qwen2.5-VL-7B-Instruct [31], Llama-3.2-3B-Instruct, Llama-3.1-8B-Instruct [1], Ministral-8B-Instruct-2410, and Mistral-7B-Instruct-v0.3 [45], to establish the generalizability of ReDit.

**Random Seeds**. Our method incorporates random sampling, which can introduce variance to the experimental outcomes. To thoroughly assess the impact of this stochasticity and ensure the robustness of our results, we executed all main experiments using multiple distinct random seeds. Specifically, we selected five seeds: None (no seed), 42, 123, 888, 2025, and 9999. All metrics reported in our final results represent the mean and variance computed across these five independent runs.

**Other Training Settings.** For parameter-efficient fine-tuning, we employed Low-Rank Adaptation (LoRA) [46]. Our implementation leverages the official GRPO implementation within the TRL library [47]. Specific configurations for LoRA and GRPO parameters are detailed in the Appendix D.3. Model evaluation was conducted using the OpenCompass [48]. All experiments were executed on one NVIDIA H20 GPU.

## 5.2 Main Results

In our main experiments, we validate the effectiveness of our proposed ReDit. For these experiments, we primarily use either a uniform smoothing kernel with radius $a = 0.05$ or a Gaussian smoothing kernel with standard deviation $\sigma = a/\sqrt{3}$. More experimental results can be found in the Appendix F.

**Accelerated Convergence Across Datasets and LLMs.** We demon-strate that integrating our proposed method, ReDit, with GRPO substantially accelerates convergence and improves final performance across a wide range of datasets (Fig. 7) and LLMs, including Llama-3.2-3B, Llama-3.1-8B, Ministral-8B and Mistral-7B (Fig. 6). On all tested models, both Gaussian and uniform variants of ReDit enable GRPO to reach a competitive performance level within merely 1000 training steps. Notably, this performance already surpasses that of the baseline GRPO trained for the full 9000 steps. Consequently, ReDit not only enhances training efficiency but also leads to

Table 2: Test mean and variance of accuracy comparison across datasets for original Backbone, GRPO, and ReDit.

| Name | GSM8K | MATH | Geometry3K |
|---|---|---|---|
| Backbone | 84.91 | 39 | 40.43 |
| GRPO(Baseline) | 89.07 | 48.01 | 43.10 |
| w/ **ours(Gauss)** | **90.76** ($\pm$ 0.06) | **52.55** ($\pm$ 0.03) | **44.67** ($\pm$ 0.03) |
| w/ **ours(Uniform)** | 90.46 ($\pm$ 0.07) | 51.96 ($\pm$ 0.06) | 44.36 ($\pm$ 0.04) |
| $\Delta$ | +1.69 | +4.54 | +1.57 |

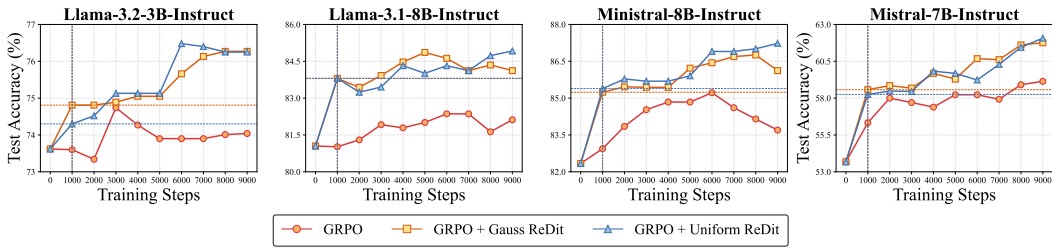

Figure 6: Accuracy of different LLMs on GSM8K. ReDit improves training efficiency and final performance in various LLMs.

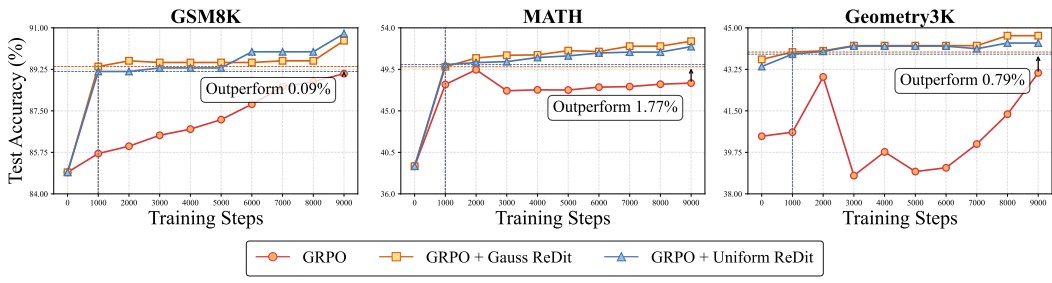

Figure 7: Test accuracy across datasets. The horizontal dashed line marks ReDit's performance at 1000 steps, which GRPO fails to match even after 9000 steps.

superior final accuracy. The Gaussian variant, in particular, consistently yields the strongest results and promotes more stable training trajectories with lower volatility compared to the baseline. We also present more experimental results in the appendix. The results on the code dataset are detailed in Appendix E.1, the results on the full parameter fine-tuning method are detailed in Appendix E.2, and the results on the Deepseek-R1 distillation model are detailed in Appendix E.3.

**Generalization to Diverse Baselines.** Fig. 5 presents results from applying ReDit to additional reinforcement learning baselines (DAPO, Dr.GRPO, and REINFORCE++) on the GSM8K dataset. Across all algorithms, ReDit (both Gaussian and uniform variants) consistently enhances performance and accelerates learning. Beyond these early-stage improvements, ReDit also substantially boosts the final accuracy of these baselines, as quantitatively demonstrated in Table 1. These accuracy gains (Table 2) complement the qualitative evidence in Fig. 5, confirming that ReDit enables faster and more stable learning across diverse algorithms.

**Optimal Performance with Scheduled Perturbation.** We further investigate convergence behavior under various scheduled perturbation schemes: SquareRoot, Cosine, and CosineReverse perturbations. These schedules dynamically adjust perturbation variance throughout training, potentially benefiting model learning. Fig. in the Appendix F.4 illustrates the different perturbation schedules, while Fig. 8 presents their performance. Compared to standard GRPO, ReDit achieves both faster convergence and superior final performance, with the CosineReverse perturbation schedule yielding particularly strong results. Additional details are provided in the Appendix F.4.

## 5.3 Ablation Studies

**Perturbation variance affects performance.** To study the sensitivity of ReDit to the perturbation amplitude, we performed an ablation study by varying the parameter $a$ in the Gaussian smoothing kernel with standard deviation $\sigma = a/\sqrt{3}$. This effectively changes the variance of the applied perturbation. As shown in Fig. 9, applying reward smoothing (i.e., for any $a > 0.00$) consistently leads to faster convergence compared to the baseline without smoothing ($a = 0.00$). Moreover, in most cases, increasing the perturbation amplitude (larger $a$) tends to improve the final performance of the model. Notably, the configuration with $a = 0.05$ shows superior performance, achieving not only

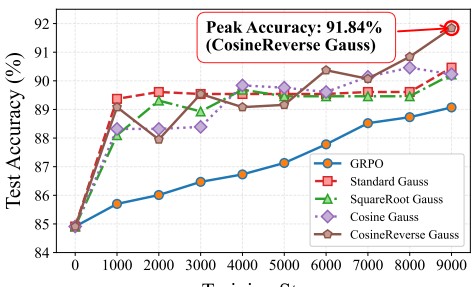

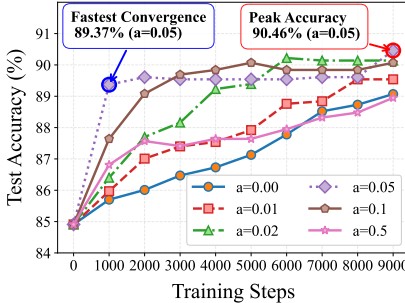

Figure 8: CosineReverse perturbation achieves the best performance.

Figure 9: Appropriate perturbation achieves the best performance.

the fastest convergence but also the best peak model performance, see the Fig. 9 annotation. However, these results highlight a key trade-off. While moderate perturbations are beneficial, excessive perturbations (e.g., $a = 0.5$) may over-smooth the reward landscape. This may mask the original reward signal and lead to performance degradation. Conversely, if the perturbation variance is too small (e.g., $a = 0.01$), the smoothing effect is small and the improvement over the baseline is limited. This suggests that there is an optimal perturbation variance. We recommend conducting preliminary experiments on a smaller dataset to effectively determine this optimal variance before applying it to larger-scale training scenarios. For a detailed theoretical introduction to $\sigma$, please refer to Section 6.

**Isolating the Effect on Discrete Rewards.** To verify that the performance gains of ReDit stem specifically from smoothing discrete rewards, we conducted a crucial ablation study. In this experiment, we replaced the discrete reward signal with a continuous one generated by a reward model pre-trained on human preference data. This model provides a continuous quality score within the range [0,1]. We then applied the ReDit perturbation mechanism directly to these continuous rewards. The results, presented in Fig. 10, show that applying ReDit in this setting yields no discernible impact on either the convergence speed of model or its final performance. This outcome strongly indicates that the benefits of ReDit are nullified when the reward landscape is already smooth. We therefore conclude that the efficacy of ReDit lies specifically in addressing the optimization challenges inherent to sparse and discrete reward signals.

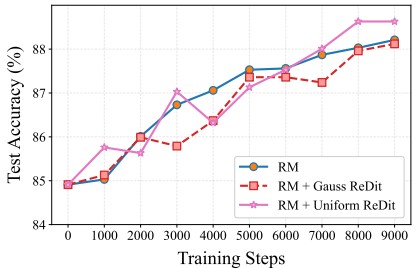

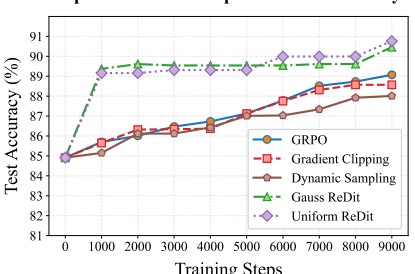

Figure 10: ReDit has little effect on improving the performance of GRPO based RM.

Figure 11: Appropriate perturbation achieves the best performance.

**Comparison with Direct Gradient Manipulation Baselines.** We benchmark ReDit against established techniques that directly address gradient instability: Gradient Clipping [49], which mitigates exploding gradients, and Dynamic Sampling [38], which alleviates vanishing gradients. The objective is to compare our ReDit approach with methods that operate directly on the gradient signal. As illustrated in Figure 11, ReDit substantially outperforms both baseline methods. We attribute this performance gap to the inherent limitations of these heuristics. Gradient Clipping, for instance, crudely truncates gradient magnitudes, a non-principled operation that can introduce significant

estimation bias. Conversely, while Dynamic Sampling can be effective for vanishing gradients, it offers no mechanism to prevent gradients from exploding. In contrast, ReDit stabilizes the training process by smoothing the reward, which provides a more principled solution to prevent both gradient vanishing and explosion, thereby leading to more efficient and effective training.

## 6 Theoretical Insights

We provides a theoretical analyzing how perturbing discrete reward signals with, e.g., Gaussian noise, accelerates RL convergence, offering a principled explanation for observed empirical benefits.

**Problem Setup.** Our analysis uses a simplified RL framework (from Eq. (1)) focusing on binary rewards $R(q, o) \in \{0, 1\}$ for complete outputs (e.g., GRPO [15]), not token-level rewards. We investigate how Gaussian noise $\epsilon \sim \mathcal{N}(0, \sigma^2)$ injection improves convergence. The perturbed objective is:

$$\tilde{J}(\pi_\theta) = \mathbb{E}_{q \sim p_Q} \left[ \mathbb{E}_{o \sim \pi_\theta(\cdot|q)} \tilde{R}(q, o) \right], \tag{6}$$

where the perturbed reward is $\tilde{R}(q, o) = R(q, o) + \epsilon$.

> **Proposition 1** (Unbiased estimate of gradient). *Introducing noise will still ensure the unbiased estimate of the gradient of the original optimization target Eq. (1), that is:*
>
> $$\mathbb{E}\left[ \nabla_\theta \tilde{J}(\pi_\theta) \right] = \mathbb{E}\left[ \nabla_\theta J(\pi_\theta) \right].$$

**Remark.** Proposition 1 provides theoretical proof that introducing Gaussian noise perturbations into the discrete reward signal preserves the unbiased nature of the policy gradient estimate. This means that, under the perturbed reward, the expected direction of the policy update is consistent with the original objective being optimized. Maintaining this unbiased nature ensures that the injected noise does not introduce systematic biases into the learning dynamics, thus providing a theoretical basis for the empirical observation that our approach helps to consistently improve performance. See Appendix B.2 for a detailed proof.

> **Proposition 2** (Introducing the variance of gradient estimation). *Suppose we are optimizing a non-degenerate strategy, that is, its gradient $\nabla_\theta \log \pi_\theta$ is not completely zero. Introducing noise will introduce gradient noise on the originally calculated gradient, and its variance is:*
>
> $$Var(Gradient\ Noise) = \sigma^2 \cdot \mathbb{E}\left[ \|\nabla_\theta \log \pi_\theta(o|q)\|^2 \right] > 0.$$

**Remark.** In Proposition 2, we analyze how Gaussian reward perturbations affect the variance of policy gradient estimates. Adding Gaussian noise $\epsilon \sim \mathcal{N}(0, \sigma^2)$ to the reward introduces a "gradient noise" component proportional to $\epsilon \cdot \nabla_\theta \log \pi_\theta(o|q)$ in the gradient estimate. The increased variance has significant optimization benefits: **Mitigate vanishing gradients:** Gradient noise provides consistent stochastic updates even when the original gradient terms are small or vanishing, thus helping to avoid flat regions. **Avoid exploding gradients:** The randomness induced by the noise enables the optimization trajectory to probabilistically bypass unstable regions of high curvature. Furthermore, the noise variance $\sigma$ can be adjusted to control the magnitude of the gradient noise for optimal results. This mechanism enhances the robustness of policy optimization and explains the empirical improvements observed in training stability and convergence speed from reward perturbations. For detailed derivation, see Appendix B.3.

## 7 Limitations and Conclusions

ReDit Improve the stability of reinforcement learning with zero-mean reward noise - theoretically smoothing gradients, preventing gradient instability, and accelerating convergence by increasing reward variance. Benchmarks verify significant improvements in convergence speed and performance. While our approach works, it requires careful tuning of the perturbation variance (although we adopt a small dataset search strategy, extensive experiments are still needed), and future work will aim to achieve automatic variance.

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

## Acknowledgement

This work was supported by the Shenzhen Science and Technology Program (Grant No. ZDSYS20220527171400002), the National Natural Science Foundation of China (Grant Nos. 62271324, 62231020, and 62371309), and the Open Research Fund from Guangdong Laboratory of Artificial Intelligence and Digital Economy (SZ) (Grant No. GML-KF-24-32).


# A Related Work

**Reinforcement Learning with Discrete Rewards.** Group Relative Policy Optimization (GRPO) [15] utilizes discrete rewards generated by a rule-based reward function to guide the policy model update. This reward function, known for its simplicity and unbiasedness, effectively mitigates reward hacking and has demonstrated strong performance. However, GRPO faces challenges related to slow training speed and unstable gradients during training. To address these issues, various methods have been proposed. DAPO [38] introduced a dynamic sampling strategy to improve gradient effectiveness by dynamically filtering invalid samples, thereby increasing sample efficiency, although this reduced training speed. CPPO [50] prunes completions with low absolute advantages, significantly reducing the number of gradient calculations and updates required, which enhances training efficiency but can lead to gradient estimation errors. GPG [51] directly optimizes the original reinforcement learning objective, eliminating the need for a proxy loss function and improving training efficiency. However, this simplification may result in a significant divergence between the actor and policy models. Dr.GRPO [39] improves token efficiency while maintaining inference performance. Despite these efforts, a critical challenge remains: these algorithms largely neglect the inherent difficulties introduced by discrete rewards during the optimization process. The oscillations caused by gradient vanishing and exploding are major contributors to the slow optimization speed. Our work specifically aims to overcome the challenges in gradient optimization that arise from using discrete rewards.

**Addressing Reward Design Challenges in LLM Reinforcement Learning.** Designing effective reward functions for identifying optimal strategies is a well-established area of research outside the context of Large Language Models (LLMs) [52, 53, 54]. However, a consensus on the optimal approach for LLM reinforcement learning has not yet been reached [55]. The RLHF framework proposed training a reward model to score LLM outputs [5]. A recurring challenge, as noted by numerous studies, is that low reward model accuracy can induce reward hacking [34, 35, 36]. Conversely, improving accuracy often reduces reward variance, which can slow down policy model convergence due to vanishing gradients [27]. Although the reward function presented in GRPO provides perfectly correct rewards, thereby avoiding reward hacking, it exacerbates gradient instability and hinders optimization speed [15, 39]. Recent theoretical findings indicate that a successful reward function requires a trade-off between variance and inaccuracy [41]. Motivated by this, our work seeks to design a reward function that effectively addresses the problem of reward hacking while simultaneously facilitating efficient optimization.

# B Theorems and proofs

## B.1 Definitions

From Definition 1, 2 in [41], The accuracy and variance of the reward function is as follows:

**Definition 1.** *Given a prompt $x \in \mathcal{X}$, the accuracy of a reward model $r_{RM} : \mathcal{X} \times \mathcal{Y} \to [-1, 1]$ with respect to a distribution $\mathcal{D}$ over unordered output pairs is defined by:*

$$Acc_{x,\mathcal{D}}(r_{RM}) := \mathbb{E}_{\{y,y'\}\sim\mathcal{D}}\left[\mathbb{1}\left[sign\big(r_{RM}(x,y) - r_{RM}(x,y')\big) = sign\big(r_G(x,y) - r_G(x,y')\big)\right]\right], \tag{7}$$

*where $r_G$ is the ground truth reward, $\mathbb{1}[\cdot]$ is an indicator function, and $sign : \mathbb{R} \to \{-1, 0, 1\}$ is the sign function.*[2]

**Definition 2.** *Given a policy $\pi_\theta$, prompt $x \in \mathcal{X}$, and reward model $r_{RM} : \mathcal{X} \times \mathcal{Y} \to [-1, 1]$, the reward variance induced by $r_{RM}$ for $\pi_\theta$ and $x$ is defined by:*

$$Var_{y\sim\pi_\theta(\cdot|x)}[r_{RM}(x,y)] := \mathbb{E}_{y\sim\pi_\theta(\cdot|x)}\left[\left(r_{RM}(x,y) - \mathbb{E}_{y'\sim\pi_\theta(\cdot|x)}\big[r_{RM}(x,y')\big]\right)^2\right]. \tag{8}$$

---

[2]For a set of prompts, accuracy refers to the mean accuracy over the set.

## B.2 Proof of Proposition 1

The proof of Proposition 1 is expressed as follows:

*Proof.* By the policy gradient theorem, the gradient of the original objective (1) expands to:

$$\nabla_\theta J(\pi_\theta) = \mathbb{E}_{q \sim p_Q} \mathbb{E}_{o \sim \pi_\theta(\cdot|q)} \left[ R(q,o) \nabla_\theta \log \pi_\theta(o|q) \right]. \tag{9}$$

For the noise-injected objective, its gradient becomes:

$$\nabla_\theta \tilde{J}(\pi_\theta) = \mathbb{E}_{q \sim p_Q} \mathbb{E}_{o \sim \pi_\theta(\cdot|q)} \left[ \tilde{R}(q,o) \nabla_\theta \log \pi_\theta(o|q) \right]. \tag{10}$$

Substituting $\tilde{R}(q,o) = R(q,o) + \epsilon$ and leveraging linearity of expectation:

$$\mathbb{E}\left[ \nabla_\theta \tilde{J} \right] = \mathbb{E}_{q,o,\epsilon} \left[ (R(q,o) + \epsilon) \nabla_\theta \log \pi_\theta(o|q) \right] \tag{11}$$

$$= \underbrace{\mathbb{E}_{q,o} \left[ R(q,o) \nabla_\theta \log \pi_\theta(o|q) \right]}_{\mathbb{E}[\nabla_\theta J]} + \mathbb{E}_\epsilon[\epsilon] \cdot \mathbb{E}_{q,o} \left[ \nabla_\theta \log \pi_\theta(o|q) \right]. \tag{12}$$

Zero-mean noise: $\mathbb{E}_\epsilon[\epsilon] = 0$ by definition of $\mathcal{N}(0, \sigma^2)$. Thus, the cross-term vanishes:

$$\mathbb{E}[\nabla_\theta \tilde{J}] = \mathbb{E}[\nabla_\theta J] + 0 = \mathbb{E}[\nabla_\theta J]. \tag{13}$$

$\square$

## B.3 Proof of Proposition 2

*Proof.* Consider the perturbed objective function with noise-augmented reward $R(q,o) + \epsilon$. The estimated value of the gradient of the noise enhancement objective function using $n$ samples is:

$$\nabla \hat{\tilde{J}}(\theta) = \frac{1}{n} \sum_{i=1}^{n} \left[ \nabla \log \pi_\theta(o_i|q_i) \cdot (R(q_i, o_i) + \epsilon_i) \right], \tag{14}$$

where $\epsilon_i \sim \mathcal{N}(0, \sigma^2)$ is the Gaussian noise. The original reward gradient is:

$$\nabla \hat{J}(\theta) = \frac{1}{n} \sum_{i=1}^{n} \left[ \nabla \log \pi_\theta(o_i|q_i) \cdot (R(q_i, o_i)) \right]. \tag{15}$$

Under this condition, the Eq. (14) simplifies to:

$$\nabla \hat{\tilde{J}}(\theta) = \underbrace{\nabla \hat{J}(\theta)}_{\text{origin gradient}} + \underbrace{\frac{1}{n} \sum_{i=1}^{n} \left[ \nabla \log \pi_\theta(o_i|q_i) \cdot \epsilon_i \right]}_{\text{noise gradient}}. \tag{16}$$

While the expectation $\mathbb{E}_\epsilon[\epsilon] = 0$ implies the noise contribution's mean is zero, the variance of the gradient term persists. To compute this variance, we use the definition: $\text{Var}(X) = \mathbb{E}[X^2] - (\mathbb{E}[X])^2$. Applying this to the noise-induced component $\epsilon \cdot \nabla_\theta \log \pi_\theta(o|q)$, we get:

$$\text{Var}\left( \epsilon \cdot \nabla_\theta \log \pi_\theta(o|q) \right) = \mathbb{E}\left[ \epsilon^2 \cdot \|\nabla_\theta \log \pi_\theta(o|q)\|^2 \right] - \left( \mathbb{E}\left[ \epsilon \cdot \nabla_\theta \log \pi_\theta(o|q) \right] \right)^2. \tag{17}$$

Since $\mathbb{E}[\epsilon] = 0$, the second term vanishes. For the first term, note that:

$$\mathbb{E}[\epsilon^2] = \text{Var}(\epsilon) + (\mathbb{E}[\epsilon])^2 = \sigma^2 + 0 = \sigma^2. \tag{18}$$

This allows us to simplify the variance expression to:

$$\text{Var(noise gradient)} = \text{Var}\left( \epsilon \cdot \nabla_\theta \log \pi_\theta \right) = \sigma^2 \cdot \mathbb{E}\left[ \|\nabla_\theta \log \pi_\theta(o|q)\|^2 \right] > 0, \tag{19}$$

provided $\nabla_\theta \log \pi_\theta$ is not identically zero (a reasonable assumption for non-degenerate policies).

$\square$

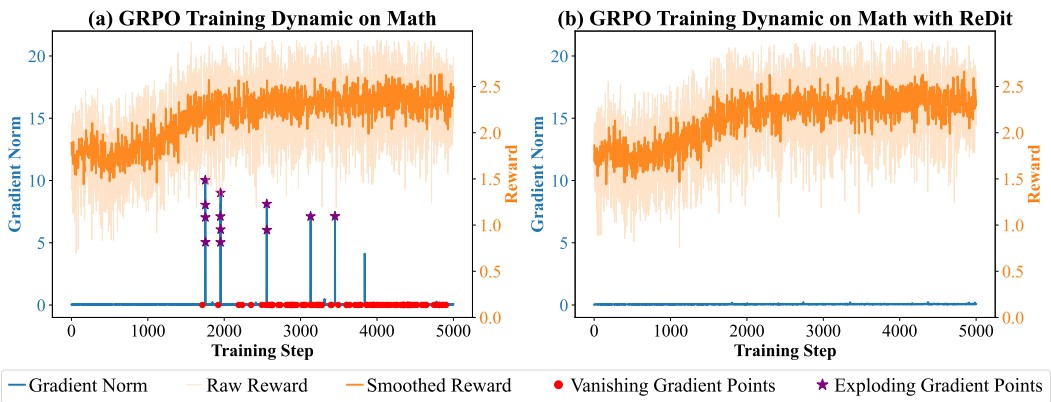

Figure 12: Training Dynamics of Gradient Norm and Reward on Math Dataset.

## C  Training Dynamic

In this section, we show more Training Dynamic information.

Figure 12 shows the training dynamics of using and not using ReDit on the Math dataset, indicating that using ReDit can solve the problems of gradient oscillation and gradient vanishing, and improve training stability

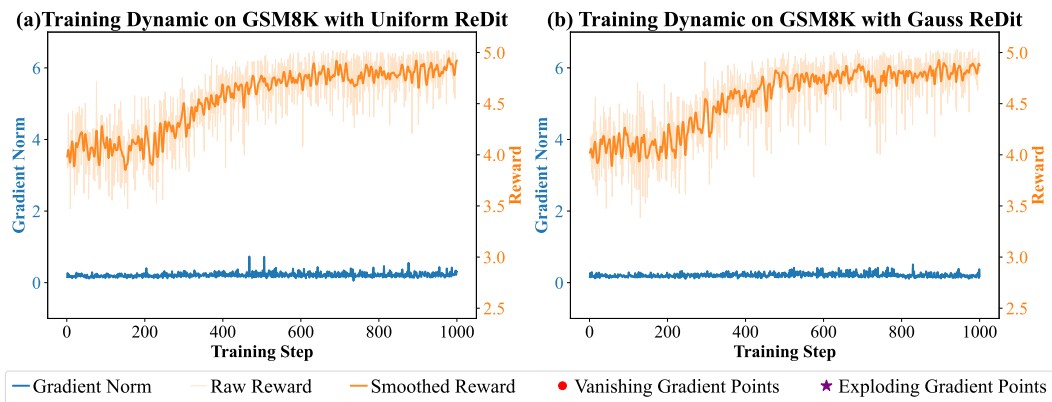

Figure 13: Training dynamics of gradient norm and reward on the GSM8K dataset, showing the impact of perturbations of different distributions.

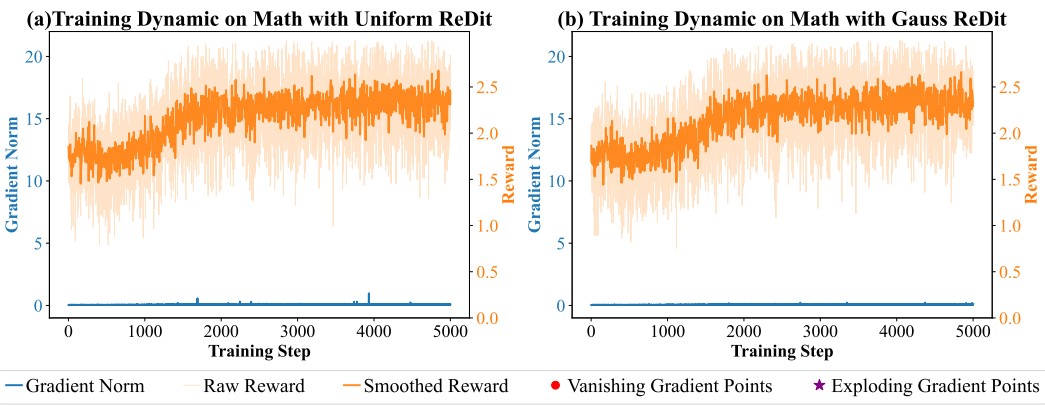

Figure 14: Training dynamics of gradient norm and reward on the Math dataset, showing the impact of perturbations of different distributions.

Fig 13 and Fig 14 Training dynamics using uniform and Gaussian perturbations. For both uniform and Gaussian perturbations, ReDit shows amazing gradient stability and training stability.

# D   Experimental setting

## D.1   Dataset

In this section, we introduce the statistics of the dataset and the additional processing performed on the dataset. The statistics of the dataset are shown in Table 3.

Table 3: Number of samples in the train, validation, and test datasets for various dateset.

| Number of samples | train dataset | validation dataset | test dataset |
|---|---|---|---|
| GSM8K | 7473 | - | 1319 |
| MATH | 7506 | - | 5003 |
| Geometry3K | 2100 | 300 | 601 |

In addition, We added new templates to the original dataset to ensure the model could complete the required tasks and output formats. It is important to note that the added templates did not alter the original dataset, and special processing was performed for different LLMs. The specific examples are as follows:

**Dataset Format of GSM8K**

```
dataset: GSM8K
    "prompt": [
        {"role": "system", "content": "Respond in the following format:
        <reasoning> ... </reasoning> <answer> ...</answer>"},
        {"role": "user", "content": "What is the largest single-digit prime number?"},
        {"role": "assistant", "content": "<reasoning> 9 is divisble by 3 and 8
        is divisible by 2, but 7 is prime. </reasoning>
        <answer>7</answer>",
        {"role": "user", "content": {question}}
        ],
    "answer": {answer}
```

**Dataset Format of MATH**

```
dataset: MATH
    "prompt": [
        {"role": "system", "content": "Respond in the following format:
        <reasoning> ... </reasoning> <answer> ...</answer>"},
        {"role": "user", "content": "{question}
        Let"s think step by step and output the final answer within \\boxed{}."
        ],
    "answer": {answer}
```

```
dataset: Geometry3K
    "prompt": [
        {"role": "user", "content": [{
            "type": "image",
            "image": {image},
        },
        {
            "type": "text",
            "text": {question} + ".
            You FIRST think about the reasoning process as an internal monologue and
            then provide the final answer. The reasoning process MUST BE enclosed
            within <think> </think> tags. The final answer MUST BE put in \\boxed{}."
            },],
        }
    ]
    "answer": {answer}
```

## D.2  Reward function

We design five reward functions for the GSM8K dataset and show how to implement ReDit:

**GSM8K Accuracy Reward Function**

```
1  def correctness_reward_func_with_noise(prompts, completions, answer
       , **kwargs) -> list[float]:
2      def extract_number(s: str) -> str:
3          match = re.search(r'\d+', s)
4          return match.group(0) if match else ''
5      responses = [completion[0]['content'] for completion in
           completions]
6      q = prompts[0][-1]['content']
7      extracted_responses = [extract_xml_answer(r) for r in responses
           ]
8      original_rewards = [2.0 if extract_number(r) == extract_number(
           a) else 0.0 for r, a in zip(extracted_responses, answer)]
9
10     # ReDit add
11     noisy_rewards = [r + random.uniform(-m * 2.0, m * 2.0) for r in
             original_rewards]
12     #noisy_rewards = [r + random.gauss(0, 2.0 * m / (3 ** 0.5)) for
             r in original_rewards]
13     return noisy_rewards
```

### GSM8K Int Reward Function

```python
def int_reward_func_with_noise(completions, **kwargs) -> list[float
    ]:
    responses = [completion[0]['content'] for completion in
        completions]
    extracted_responses = [extract_xml_answer(r) for r in responses
        ]
    original_rewards = [0.5 if r.isdigit() else 0.0 for r in
        extracted_responses]

    # ReDit add
    noisy_rewards = [r + random.uniform(-m * 0.5, m * 0.5) for r in
        original_rewards]
    #noisy_rewards = [r + random.gauss(0, 0.5 * m / (3 ** 0.5)) for
        r in original_rewards]
    return noisy_rewards
```

### GSM8K Strict Format Reward Function

```python
def strict_format_reward_func_with_noise(completions, **kwargs) ->
    list[float]:
    pattern = r"^<reasoning>\n[\s\S]*?\n</reasoning>\n<answer>\n[\s
        \S]*?</answer>$"
    completion_contents = [completion[0]["content"].strip() for
        completion in completions]
    matches = [re.match(pattern, content, re.DOTALL | re.MULTILINE)
        for content in completion_contents]
    original_rewards = [1.0 if match else 0.0 for match in matches]

    # ReDit add
    noisy_rewards = [r + random.uniform(-m * 1.0, m * 1.0) for r in
        original_rewards]
    #noisy_rewards = [r + random.gauss(0, 1.0 * m / (3 ** 0.5)) for
        r in original_rewards]
    return noisy_rewards
```

### GSM8K Sort Format Reward Function

```python
def soft_format_reward_func_with_noise(completions, **kwargs) ->
    list[float]:
    pattern = r"^<reasoning>[\s\S]*?</reasoning>[\s\S]*?<answer>[\s
        \S]*?</answer>$"
    completion_contents = [completion[0]["content"].strip() for
        completion in completions]
    matches = [re.match(pattern, content, re.DOTALL | re.MULTILINE)
        for content in completion_contents]
    original_rewards = [1.0 if match else 0.0 for match in matches]

    #  ReDit add
    noisy_rewards = [r + random.uniform(-m * 1.0, m * 1.0) for r in
        original_rewards]
    #noisy_rewards = [r + random.gauss(0, 1.0 * m / (3 ** 0.5)) for
        r in original_rewards]
    return noisy_rewards
```

```python
def xmlcount_reward_func_with_noise(completions, **kwargs) -> list[
    float]:
    def count_xml(text) -> float:
        count = 0.0
        if text.count("<reasoning>\n") == 1:
            count += 0.125
        if text.count("\n</reasoning>\n") == 1:
            count += 0.125
        if text.count("\n<answer>\n") == 1:
            count += 0.125
            #count -= len(text.split("\n</answer>\n")[-1])*0.001
        if text.count("\n</answer>") == 1:
            count += 0.125
            count -= (len(text.split("\n</answer>")[-1]) - 1)*0.001
        return count
    contents = [completion[0]["content"] for completion in
        completions]
    original_rewards = [count_xml(c) for c in contents]

    # ReDit add
    noisy_rewards = [r + random.uniform(-m * 0.5, m * 0.5) for r in
        original_rewards]
    #noisy_rewards = [r + random.gauss(0, 0.5 * m / (3 ** 0.5))
        for r in original_rewards]
    return noisy_rewards
```

As shown in the above code block, ReDit does not need to be modified in a complex way, only the reward function needs to be modified, and any method can be easily integrated. The reward functions of other datasets can be found in the code.

### D.3 Specific experimental parameters

In this section, we present the experimental parameters, including LoRA parameters, GRPO and other baseline experimental parameters.

Table 4: LoRA Parameters

| LoRA Target | LoRA Rank | LoRA Alpha | LoRA Dropout |
|---|---|---|---|
| q & v Proj | 8 | 64 | 0.05 |

Table 5: GRPO Parameters

| Learning Rate | Num Generations | Epochs |
|---|---|---|
| 5e-6 | 4 | 10 |

Table 6: DAPO Parameters

| Clip Ratio Low | Clip Ratio Low | Clip Ratio C | Num Generations Max |
|---|---|---|---|
| 0.2 | 0.28 | 10.0 | 10 |

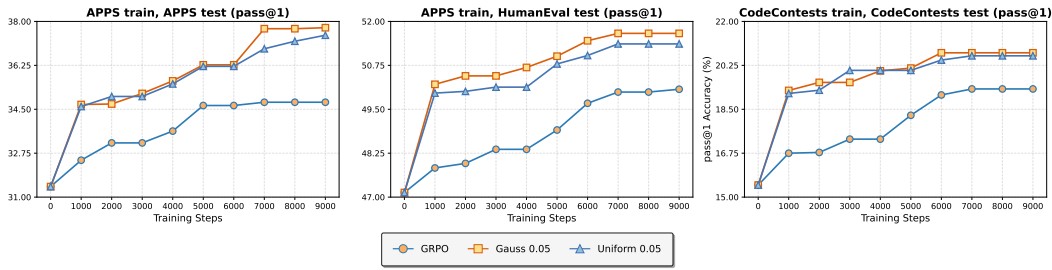

Figure 15: Performance comparison on three code generation benchmarks: (left) APPS test, (center) HumanEval test, and (right) CodeContests test. The $pass@1$ accuracy is reported across training steps. Both ReDit variants (Gauss 0.05 and Uniform 0.05) consistently and significantly outperform the GRPO baseline, confirming the general applicability of our method to the coding domain.

# E    Additional experiments

## E.1    Results on the Code Generation Datasets

To validate the general applicability of our method (ReDit) beyond mathematical reasoning, we conducted a comprehensive evaluation on the domain of code generation. We performed experiments on three widely-used coding benchmarks: APPS, HumanEval, and CodeContests. This evaluation was designed to test the hypothesis that the core benefit of ReDit—stabilizing the learning signal to improve optimization—is a general principle that is not limited to a single domain.

The results are presented in 15. The plots consistently demonstrate that both the Gaussian (Gauss 0.05) and Uniform (Uniform 0.05) variants of ReDit significantly and consistently outperform the GRPO baseline across all three coding benchmarks. On all datasets, our method not only achieves a higher final $pass@1$ accuracy but also exhibits a faster convergence rate. This strongly suggests that ReDit provides a more robust optimization pathway, and its benefits generalize effectively to complex tasks such as code generation.

## E.2    Results on Full Parameter Fine-Tuning

To confirm that the benefits of ReDit are not limited to parameter-efficient fine-tuning (PEFT) methods like LoRA, we conducted additional experiments using a full parameter fine-tuning approach. This evaluation addresses whether ReDit's effectiveness is a general property of the optimization process itself, rather than an artifact of a specific tuning method [56].

For these experiments, we utilized a setup that differs from our primary PEFT experiments; specifically, we employed the VERL framework for training with 8 GPUs. We evaluated this full fine-tuning setup on our three mathematical reasoning benchmarks: GSM8K, MATH, and Geo3k.

The results are presented in 16. The plots clearly demonstrate that ReDit (both Gauss 0.05 and Uniform 0.05 variants) consistently outperforms the GRPO baseline across all three benchmarks in this demanding full-tuning setting. The performance gap is particularly notable on the MATH and Geo3k datasets, where the GRPO baseline shows signs of stagnation, while our method continues to improve. These findings confirm that ReDit is a robust and general-purpose technique, delivering consistent performance gains in both parameter-efficient and full fine-tuning paradigms.

## E.3    Results on DeepSeek Distillation Models

A potential concern regarding our primary results is that they are predominantly focused on the Qwen2.5 model family. To further demonstrate the robustness and architectural generalizability of ReDit, we conducted additional experiments to validate its effectiveness on specialized reasoning models.

Specifically, we evaluated our method on two expert distillation models: **DeepSeek-R1-Distill-Llama-8B** and **DeepSeek-R1-Distill-Qwen-7B**. We used the GSM8K benchmark to compare their

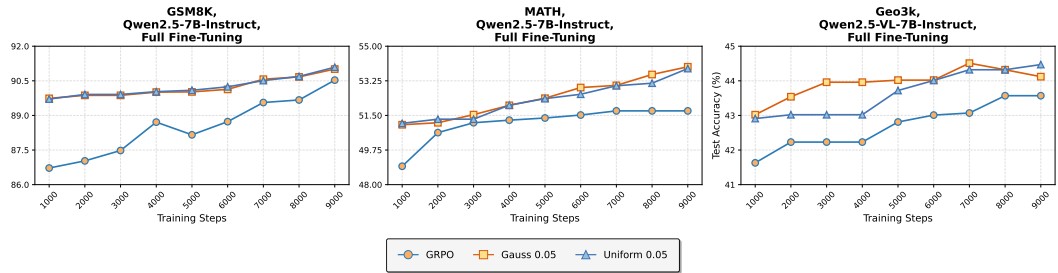

Figure 16: Performance comparison on mathematical reasoning benchmarks using **full parameter fine-tuning**. The plots show test accuracy across training steps for (left) GSM8K, (center) MATH, and (right) Geo3k. In this full-tuning setting, both ReDit variants (Gauss 0.05 and Uniform 0.05) consistently achieve higher test accuracy than the GRPO baseline, confirming that our method's benefits generalize beyond parameter-efficient tuning (PEFT).

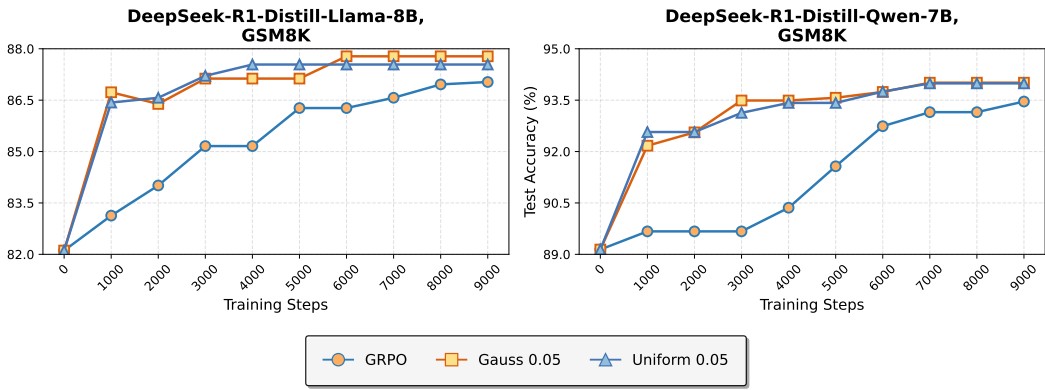

Figure 17: Evaluating ReDit's generalizability on specialized reasoning models. The plots show Test Accuracy (%) on the GSM8K benchmark for (left) **DeepSeek-R1-Distill-Llama-8B** and (right) **DeepSeek-R1-Distill-Qwen-7B**. These results confirm that ReDit's performance advantage over the GRPO baseline holds across different model architectures, not just the Qwen models used in the main experiments.

mathematical reasoning performance against the GRPO baseline. The results are presented in 17. The plots clearly show that both the Gaussian (Gauss 0.05) and Uniform (Uniform 0.05) variants of ReDit consistently outperform the GRPO baseline on both DeepSeek models.

This finding is significant as it confirms that ReDit's ability to stabilize the training signal and improve performance is a general principle. It is not limited to a single model family but holds true across various model architectures, including those specifically optimized for reasoning tasks.

# F   More result

In this section, we present detailed numerical results for all experiments.

## F.1   Main Result

In this section, we show the results in Figure 7, the performance of GRPO and GRPO+ReDit on different datasets.

Tables  7,  8, 9 show the comparison of ReDit on different datasets. ReDit significantly improves the convergence speed of GRPO. At any same step, ReDit achieves better performance.

Table 7: Performance Comparison of Different Training Steps on the Math Dataset

| Method \Step | 0 | 1000 | 2000 | 3000 | 4000 | 5000 | 6000 | 7000 | 8000 | 9000 |
|---|---|---|---|---|---|---|---|---|---|---|
| Instruct model | 39 | - | - | - | - | - | - | - | - | - |
| GRPO | - | 47.86 | 49.46 | 47.18 | 47.28 | 47.26 | 47.57 | 47.63 | 47.89 | 48.01 |
| Uniform ReDit | - | 50.02 | 50.23 | 50.34 | 50.78 | 50.96 | 51.27 | 51.37 | 51.37 | 51.96 |
| Gauss ReDit | - | 49.78 | 50.73 | 51.03 | 51.07 | 51.53 | 51.43 | 52.01 | 52.01 | 52.55 |

Table 8: Performance Comparison of Different Training Steps on the GSM8K Dataset

| Method \Step | 0 | 1000 | 2000 | 3000 | 4000 | 5000 | 6000 | 7000 | 8000 | 9000 |
|---|---|---|---|---|---|---|---|---|---|---|
| Instruct model | 84.91 | - | - | - | - | - | - | - | - | - |
| GRPO | - | 85.70 | 86.01 | 86.47 | 86.73 | 87.13 | 87.78 | 88.52 | 88.73 | 89.07 |
| Uniform ReDit | - | 89.16 | 89.16 | 89.31 | 89.31 | 89.31 | 89.99 | 89.99 | 89.99 | 90.76 |
| Gauss ReDit | - | 89.02 | 89.37 | 89.61 | 89.54 | 89.54 | 89.54 | 89.61 | 89.61 | 90.46 |

## F.2 Baseline Result

In this section, we present all numerical results in Fig. 5. As shown in Table 10, we demonstrate the effect of using ReDit on GSM8K based on the GRPO improvement method. The experimental results show that ReDit can also improve the convergence speed and performance on these algorithms.

## F.3 Variance Result

In this section, we show more results on the performance of ReDit as the perturbation changes. As shown in Figure 18, the variance of uniform perturbation is similar to the variance of Gaussian perturbation, and the appropriate variance can achieve the best performance. The specific numerical results are shown in Tables 11 and 12.

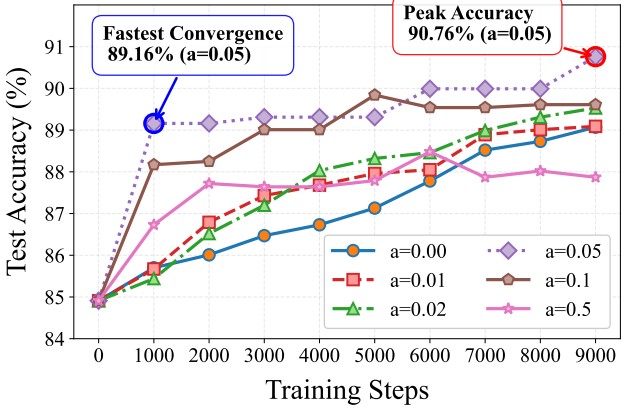

Figure 18: ReDit uniform perturbation performance changes with variance.

## F.4 Scheduled Perturbation Result

In this section, we show the changing trends of different scheduled perturbation strategies, as shown in Figure 19. We took the perturbation of Gauss distribution as an example and conducted experiments. The experimental results are shown in Table 13. The CosineReverse strategy shows the best performance.

Table 9: Performance Comparison of Different Training Steps on the Geometry3K Dataset

| Method \Step | 0 | 1000 | 2000 | 3000 | 4000 | 5000 | 6000 | 7000 | 8000 | 9000 |
|---|---|---|---|---|---|---|---|---|---|---|
| Instruct model | 40.43 | - | - | - | - | - | - | - | - | - |
| GRPO | - | 40.60 | 42.93 | 38.77 | 39.77 | 38.94 | 39.10 | 40.10 | 41.36 | 43.10 |
| Uniform ReDit | - | 43.37 | 43.89 | 44.01 | 44.23 | 44.23 | 44.23 | 44.12 | 44.36 | 44.36 |
| Gauss ReDit | - | 43.67 | 43.98 | 44.03 | 44.25 | 44.25 | 44.25 | 44.25 | 44.67 | 44.67 |

Table 10: Performance Comparison at Different Training Steps on Different Baseline

| Method \Step | 1000 | 2000 | 3000 | 4000 | 5000 | 6000 | 7000 | 8000 | 9000 |
|---|---|---|---|---|---|---|---|---|---|
| DAPO | 84.99 | 86.20 | 86.35 | 86.35 | 86.75 | 87.04 | 87.12 | 87.17 | 87.52 |
| Uniform ReDit | 87.03 | 87.15 | 87.26 | 87.54 | 87.54 | 87.69 | 87.83 | 88.03 | 88.57 |
| Gauss ReDit | 87.76 | 87.96 | 88.01 | 88.01 | 88.10 | 88.37 | 88.67 | 88.96 | 89.34 |
| DR.GRPO | 84.69 | 84.23 | 84.53 | 84.91 | 85.67 | 85.67 | 85.67 | 85.90 | 86.13 |
| Uniform ReDit | 86.27 | 86.36 | 86.45 | 86.54 | 86.75 | 87.03 | 87.26 | 87.16 | 87.34 |
| Gauss ReDit | 86.47 | 86.23 | 87.10 | 87.16 | 87.56 | 87.67 | 87.67 | 87.67 | 87.69 |
| REINFORCE++ | 84.91 | 84.69 | 85.06 | 85.14 | 85.14 | 85.14 | 86.10 | 86.17 | 86.25 |
| Uniform ReDit | 86.21 | 86.11 | 86.67 | 86.31 | 86.75 | 87.01 | 87.26 | 87.59 | 87.59 |
| Gauss ReDit | 86.17 | 86.27 | 86.47 | 86.83 | 86.83 | 87.06 | 87.63 | 87.76 | 87.96 |

Table 11: Performance Comparison of Different variance on the Gauss Perturbation

| Variance \Step | 1000 | 2000 | 3000 | 4000 | 5000 | 6000 | 7000 | 8000 | 9000 |
|---|---|---|---|---|---|---|---|---|---|
| 0.01 | 85.97 | 87.01 | 87.40 | 87.54 | 87.92 | 88.76 | 88.84 | 89.54 | 89.54 |
| 0.02 | 86.40 | 87.70 | 88.16 | 89.23 | 89.39 | 90.22 | 90.14 | 90.14 | 90.14 |
| 0.05 | 89.02 | 89.37 | 89.61 | 89.54 | 89.54 | 89.54 | 89.61 | 89.61 | 90.46 |
| 0.1 | 87.64 | 89.08 | 89.69 | 89.84 | 90.07 | 89.84 | 89.84 | 89.84 | 90.07 |
| 0.3 | 87.87 | 88.48 | 88.78 | 88.93 | 89.39 | 89.39 | 89.39 | 89.46 | 89.46 |
| 0.5 | 86.81 | 87.57 | 87.41 | 87.64 | 87.64 | 87.95 | 88.32 | 88.48 | 88.95 |

Table 12: Performance Comparison of Different variance on the Uniform Perturbation

| Variance \Step | 1000 | 2000 | 3000 | 4000 | 5000 | 6000 | 7000 | 8000 | 9000 |
|---|---|---|---|---|---|---|---|---|---|
| 0.01 | 85.67 | 86.79 | 87.43 | 87.68 | 87.96 | 88.05 | 88.89 | 89.01 | 89.09 |
| 0.02 | 85.44 | 86.52 | 87.20 | 88.03 | 88.32 | 88.46 | 88.99 | 89.31 | 89.53 |
| 0.05 | 89.16 | 89.16 | 89.31 | 89.31 | 89.31 | 89.99 | 89.99 | 89.99 | 90.76 |
| 0.1 | 88.17 | 88.25 | 89.01 | 89.01 | 89.84 | 89.54 | 89.54 | 89.61 | 89.61 |
| 0.3 | 87.49 | 88.25 | 88.25 | 88.02 | 88.17 | 87.95 | 88.93 | 88.70 | 88.78 |
| 0.5 | 86.73 | 87.72 | 87.64 | 87.64 | 87.79 | 88.48 | 87.87 | 88.02 | 87.87 |

Table 13: Performance Comparison of Different Scheduled Perturbation Methods

| Method \Step | 1000 | 2000 | 3000 | 4000 | 5000 | 6000 | 7000 | 8000 | 9000 |
|---|---|---|---|---|---|---|---|---|---|
| SquareRoot | 88.10 | 89.31 | 88.93 | 89.69 | 89.46 | 89.46 | 89.46 | 89.46 | 90.22 |
| SquareRootReverse | 88.55 | 89.54 | 89.46 | 90.07 | 90.07 | 89.31 | 89.61 | 89.54 | 89.69 |
| Factor | 88.25 | 88.63 | 89.69 | 89.46 | 89.23 | 89.54 | 89.46 | 89.31 | 89.69 |
| FactorReverse | 88.48 | 88.32 | 89.39 | 88.78 | 88.93 | 89.54 | 89.61 | 89.76 | 89.46 |
| MutilFactor | 87.87 | 89.31 | 89.01 | 89.01 | 89.01 | 89.61 | 89.16 | 89.61 | 89.46 |
| MutilFactorReverse | 88.17 | 88.78 | 88.86 | 89.01 | 88.93 | 88.93 | 89.39 | 89.16 | 89.54 |
| Cosine | 88.32 | 88.32 | 89.39 | 89.84 | 89.76 | 89.61 | 90.14 | 90.46 | 90.23 |
| CosineReverse | 89.08 | 87.95 | 89.54 | 89.08 | 89.16 | 90.37 | 90.07 | 90.84 | 91.84 |

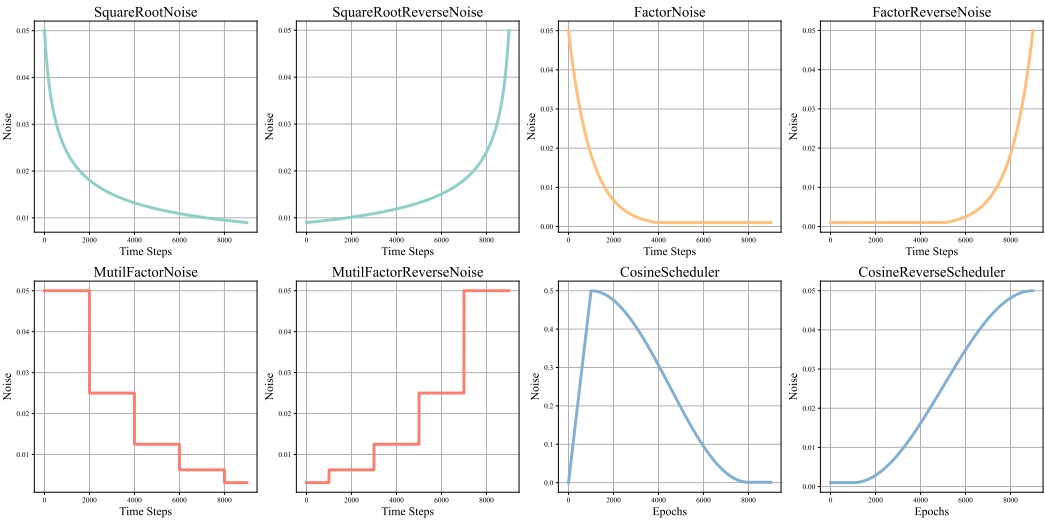

Figure 19: ReDit scheduled perturbation Variance trend with training step (taking the original variance as 0.05 as an example)

