# OpenReview forum: "ReDit: Reward Dithering for Improved LLM Policy Optimization"
_NeurIPS.cc/2025/Conference — NeurIPS 2025 poster_

### Official Review · Reviewer_UG9k · 2025-06-01

**Clarity:** 3
**Significance:** 3
**Originality:** 2
**Rating:** 5
**Confidence:** 4

**Summary:**

Proposes a reward smoothing technique, called ReDit, for improving the optimization stability of policy gradient algorithms such as GRPO, in the context of training language models to solve tasks using a discrete reward (e.g., a reward of one if the answer is correct and zero otherwise). ReDit simply adds zero-mean noise, with a fixed standard deviation, to the rewards of outputs. The motivation behind ReDit is that sufficient reward variance has been shown in prior work to be important for efficient optimization.

Experiments across datasets and policy gradient methods demonstrate that ReDit leads to improved performance and training stability. Theoretical insights are provided as to why the introduced noise accelerates optimization.

**Questions:**

--

**Ethical Concerns:**

["NO or VERY MINOR ethics concerns only"]

**Final Justification:**

The initial review raised three main concerns: lack of error bars or a measure of statistical significance in the experiments, the proof of Proposition 3 was incorrect, and some clarity issues. During the discussion period the authors have addressed all three concerns by providing additional experiments, specifying how they would remove Proposition 3 and replace it with intuitive arguments based on other parts of their theoretical analysis (leaving full formalization to future work), and detailing changes to improve clarity.

As a result, I am happy to update my assessment and recommend acceptance. The proposed method is simple, yet demonstrates promising results, and so can be of potential interest to the community.

**Limitations:**

Yes

**Paper Formatting Concerns:**

--

**Quality:**

2

**Strengths And Weaknesses:**

Strengths
---
1. Reads relatively well and is easy to follow. Provides sufficient motivation for the proposed method, based both on an empirical analysis and existing theory on the connection between reward variance and policy gradient optimization.

2. The proposed method is simple, yet demonstrates promising results, in which case I view the simplicity as a strength. Ablations over different noise schedules and standard deviations are also useful.


Weaknesses
---
1. The empirical evaluation lacks error bars or a measure of statistical significance over random seeds (e.g., standard deviation). This makes it difficult to assess whether the improvement brought forth by ReDit is consistent and significant. I strongly suggest that the authors run additional seeds and include the results. This is particularly important with policy gradient methods, which are notorious for their instability (cf. [1]).

&nbsp;

2. The proof of Proposition 3 is incorrect, and I do not believe that Proposition 3 is true as currently stated.
    - For the upper bound on $t_\gamma$, the result in [37] specifies that there exist inaccurate reward models that lead to fast optimization, and not that for every inaccurate reward model this holds. Thus, it is not true that in general when applying noise to the reward $t_\gamma$ will be bounded as specified in Proposition 3.
    - For the lower bound on $t_\gamma$, Theorem 1 in [37] considers a deterministic reward function and cannot be invoked as is. If the reward is perturbed with noise $\epsilon$, then the expected objective becomes $J(\pi_\theta) = E_{q \sim p_Q} [E_{o \sim \pi_\theta (\cdot | q)} [ E_\epsilon [\tilde{R} (q, o)] ]]$, where $\tilde{R} (q, o) = R(q, o) + \epsilon$. Following the same arguments in Theorem 1 of [37] would lead to an analogous lower bound, but that depends on $var_{o \sim \pi_\theta (\cdot | q)} ( E_{\epsilon} [\tilde{R} (q, o)] )$ as opposed to $var_{o \sim \pi_\theta (\cdot | q)} (R(q,o))$. Since the introduced noise $\epsilon$ is zero-mean, we have that $E_{\epsilon} [\tilde{R} (q, o)] = R(q, o)$, and so $var_{o \sim \pi_\theta (\cdot | q)} ( E_{\epsilon} [\tilde{R} (q, o)] ) = var_{o \sim \pi_\theta (\cdot | q)} (R(q,o))$. That is, the lower bound will not depend on the noise $\epsilon$.

    Note that the discrepancy above stems from the fact that the analysis of [37] considers optimizing the expected objective with exact gradients, as opposed to using batch-based estimates. The expected gradients are not modified by the zero-mean noise, while as Proposition 2 in this paper shows, the batch gradients are. In my view, a potential explanation for the benefit of ReDit is that, intuitively, it can help smooth out batch gradient estimates, especially when the expected gradient is small due to low reward variance, in which case the gradient estimates can be unstable.

&nbsp;

3. There are quite a few places where the clarity of the presented results can be improved. In particular, as detailed below in the additional comments part of the review, notation is used without proper introduction and there are details missing regarding what is plotted in the figures. Though, these issues should be amendable rather easily.

&nbsp;

Overall, I believe that the paper has potential due to its clarity and the promising empirical results. However, the lack of error bars or standard deviation across random seeds in the experiments (in particular, Tables 1 and 2 and Figures 5 and 6) and the incorrect Proposition 3 prevent me from recommending acceptance. If the authors are able to provide results with additional random seeds to solidify the significance of their approach and either correct or remove Proposition 3, I will gladly reassess and consider raising my rating.


[1] Henderson, P., et al., Deep reinforcement learning that matters, AAAI 2018.



Additional comments
---
1. In lines 49 and 81, the citation to [32] seems incorrect as it does not discuss the connection between reward variance and optimization.

2. Regarding Figure 1:
    - Why do the rewards not start from similar values? Since it is the same dataset and base model I would expect them to start from roughly the same value.
    - Is the reward reported for ReDit with or without the noise? For fair comparison, the results should also include the reward without noise (if this is not already the case), whether in the same plot or another.
    - What is the scale of the rewards? Without further information, I would expect them to be 0 for incorrect and 1 for correct responses, but the rewards in the figure are between 2 and 5.

3. In the red box of Theorem 2: What is a “hint”? Should that just be “prompt”?

4. In the red box of Theorem 2: The text around what the result says is imprecise. From looking at the corresponding theorem in [37], it shows that there exists a perfectly accurate reward model and a relatively inaccurate model for which the bounds on $t_\gamma$ hold.

5. More broadly, regarding the red boxes of Theorems 1 and 2, the notation and which training algorithm is considered is not properly introduced. For example, what are $f_\theta$, $S$, and $\mathcal{X}$? It is possible to look at the original theorems from [37], but it is worth being self-contained. Alternatively, since these boxes refer to results from prior work, whose exact technical setting may not be critical for the current paper, it may be worth converting the boxes to being more informal, including the main takeaway while referring to the original theorems in [37] or an appendix for the full details.

6. In line 167: $r_{RM}$ is not used elsewhere as far as I can see, so I would recommend dropping that unnecessary notation, especially given that the rewards in this paper are not given by a reward model.

7. Typo in line 138: “Eplosion” -> “Explosion”.

8. Typo in line 195: There is no line 8 in Algorithm 1. Perhaps this should have been line 5?

9. Figure 6 is missing a legend. Currently, it is not clear which line corresponds to which method. The red line seems to correspond to GRPO, but what are the remaining two lines? Are they GRPO combined with ReDit for either Gaussian or uniform noise?

10. Figures and tables do not appear in the order that they are referenced, which can be confusing. For example, Table 2 and Figure 6 are referenced before Table 1 and Figure 5.

11. Typo in line 283: “We provides a theoretical analyzing how…” is not grammatically correct.

12. Equation 5 is missing an expectation over the introduced noise $\epsilon$, which does appear in the proof of Proposition 1.

13. It is worth explicitly specifying what the variance of “Gradient Noise” refers to in Equation 7.

14. The statements in lines 302 to 305 on why the additional gradient noise is beneficial should either be hedged or refer to evidence in the literature. Currently, they are phrased as concrete truths, while they seem to be more on the line of intuitive arguments.

---

> ### Author Rebuttal · Authors · 2025-07-31
>
> We sincerely thank the reviewers for their insightful feedback. We are grateful for their recognition of our approach's clarity and simplicity, its grounding in both empirical and theoretical motivations, and the utility of our ablation study. We address your concerns below.
> # Responses to Weaknesses 1
> Thank you reviewer for this insightful and crucial suggestion. To address this concern, in addition to the initial experiments, we conducted extensive new experiments, running our approach (ReDit) and all baselines on 5 different random seeds (`42`, `123`, `888`, `2025`, `9999`), and `None` means no random seed is set. As you suggested, in the revised manuscript, all tables will be updated with the mean and variance, and the accuracy plots will be updated with error bars.  We alse show below detailed results for our `ReDit (Gauss 0.05)` approach on all seeds and baselines.
>
> ## **Setting: gsm8k, gauss,0.05**
>
> | seed/acc     | 1000    | 2000    | 3000    | 4000    | 5000    | 6000    | 7000    | 8000    | 9000    |
> |--------------|---------|---------|---------|---------|---------|---------|---------|---------|---------|
> | None         | 89.02   | 89.37   | 89.61   | 89.54   | 89.54   | 89.54   | 89.61   | 89.61   | 90.46   |
> | 42           | 89.07   | 89.35   | 89.60   | 89.53   | 89.53   | 89.36   | 89.60   | 89.69   | 90.43   |
> | 123          | 89.01   | 89.36   | 89.63   | 89.51   | 89.51   | 89.47   | 89.67   | 89.64   | 90.48   |
> | 888          | 89.02   | 89.27   | 89.69   | 89.56   | 89.52   | 89.67   | 89.65   | 89.79   | 90.39   |
> | 2025         | 88.97   | 89.39   | 89.61   | 89.59   | 89.59   | 89.65   | 89.63   | 89.72   | 90.35   |
> | 9999         | 89.08   | 89.31   | 89.58   | 89.49   | 89.60   | 89.73   | 89.66   | 89.53   | 90.51   |
> | **mean** | **89.03** | **89.34** | **89.62** | **89.54** | **89.55** | **89.57** | **89.64** | **89.66** | **90.44** |
> | **std** | **0.04** | **0.04** | **0.04** | **0.03** | **0.04** | **0.13** | **0.03** | **0.09** | **0.06** |
> ## **Setting: math, gauss,0.05**
> | seed/acc     | 1000    | 2000    | 3000    | 4000    | 5000    | 6000    | 7000    | 8000    | 9000    |
> |--------------|---------|---------|---------|---------|---------|---------|---------|---------|---------|
> | None         | 49.78   | 50.73   | 51.03   | 51.07   | 51.53   | 51.43   | 52.01   | 52.01   | 52.55   |
> | 42           | 49.73   | 50.71   | 51.04   | 51.09   | 51.53   | 51.42   | 52.03   | 52.03   | 52.59   |
> | 123          | 49.71   | 50.69   | 51.07   | 51.02   | 51.57   | 51.37   | 52.09   | 52.09   | 52.51   |
> | 888          | 49.76   | 50.67   | 51.13   | 51.03   | 51.51   | 51.25   | 52.00   | 52.05   | 52.53   |
> | 2025         | 49.73   | 50.74   | 50.96   | 50.99   | 51.46   | 51.39   | 52.08   | 51.95   | 52.56   |
> | 9999         | 49.70   | 50.79   | 50.93   | 51.02   | 51.37   | 51.47   | 52.06   | 52.06   | 52.58   |
> | **mean** | **49.74** | **50.72** | **51.03** | **51.04** | **51.50** | **51.39** | **52.05** | **52.03** | **52.55** |
> | **std** | **0.03** | **0.04** | **0.07** | **0.03** | **0.07** | **0.08** | **0.04** | **0.05** | **0.03** |
> ## **Setting: Geo3k, gauss,0.05**
> | seed/acc     | 1000    | 2000    | 3000    | 4000    | 5000    | 6000    | 7000    | 8000    | 9000    |
> |--------------|---------|---------|---------|---------|---------|---------|---------|---------|---------|
> | None         | 43.67   | 43.98   | 44.03   | 44.25   | 44.25   | 44.25   | 44.25   | 44.67   | 44.67   |
> | 42           | 43.62   | 43.91   | 44.01   | 44.22   | 44.22   | 44.23   | 44.22   | 44.62   | 44.62   |
> | 123          | 43.68   | 43.95   | 44.05   | 44.24   | 44.24   | 44.22   | 44.27   | 44.68   | 44.68   |
> | 888          | 43.69   | 43.99   | 44.02   | 44.25   | 44.23   | 44.29   | 44.25   | 44.61   | 44.62   |
> | 2025         | 43.61   | 43.89   | 44.03   | 44.29   | 44.29   | 44.29   | 44.22   | 44.63   | 44.63   |
> | 9999         | 43.69   | 44.03   | 44.09   | 44.21   | 44.21   | 44.31   | 44.31   | 44.67   | 44.69   |
> | **mean** | **43.66** | **43.96** | **44.04** | **44.24** | **44.24** | **44.27** | **44.25** | **44.65** | **44.65** |
> | **std** | **0.04** | **0.05** | **0.03** | **0.03** | **0.03** | **0.04** | **0.03** | **0.03** | **0.03** |
>
> As these tables show, the standard deviation across different runs is consistently very low (typically ≤ 0.1). This provides strong empirical evidence that ReDit's performance gains are stable, consistent, and not an artifact of a specific random initialization. These new results confirm that ReDit robustly outperforms the baselines.
> # Responses to Weaknesses 2
> We sincerely thank the reviewer for this detailed and insightful critique of our theoretical analysis in Proposition 3. Your careful examination has helped us identify an error in our original proof and allows us to clarify our argument. We agree that our initial proposition had issues and will be revised. We address the upper and lower bounds separately below.
> ## On the Upper Bound (We agree with the reviewer)
> Regarding the upper bound, you are entirely correct. We apologize for this incorrect derivation. We erroneously extrapolated the result from [37], which specifies that *there exist* certain inaccurate reward models that lead to faster optimization. We cannot prove that our specific noisy reward model, $r_{RM} = r + \epsilon$, satisfies this property in the general case.
> Action: We will remove the claim regarding the upper bound from Proposition 3 in the final version of the manuscript.
> ## On the Lower Bound (A clarification on the objective function)
> Regarding the lower bound, we would like to point out that we believe there is a misunderstanding in the understanding of the objective function in the theoretical framework of [37] and appreciate the opportunity to clarify.
> We apologize for not emphasizing in the paper that in the theoretical analysis (whether Proposition 1, Proposition 2 or Proposition 3), we are consistent with the assumptions in [37] and all theoretical analysis considers using exact gradients to optimize the expected objective function. The impact of ReDit on the gradient is detailed in our Propositions 1 and 2. As shown above, ReDit does not change the expectation of the gradient (i.e., the estimator is unbiased), but it significantly affects the variance of the gradient, which is directly related to the noise variance $\sigma^2$. Based on this theoretical setting, let us revisit the objective function being optimized. Following the approach of [37], the policy $\pi_\theta$ is updated to maximize the expected reward given the reward model $r_{RM}$:
>
> $
> \tilde{J}(\pi\_\theta) = \mathbb{E}\_{q \sim p\_Q} \left[ \mathbb{E}\_{o \sim \pi\_\theta(\cdot|q)}r\_{RM}(q, o) \right]
> $
>
> In our approach, the reward model is the noisy reward, where $r_{RM} = r + \epsilon$. This corresponds to $\tilde{R}$ in Equation (5) of our paper.
> The key lies in the application of Theorem 1 in [37]. Theorem 1 gives a lower bound on the optimization result of $\tilde{J}(\pi_\theta)$ based on the properties of the reward model $r_{RM}$ and its relationship to the true reward $r$. In our application, $r_{RM}$ is the reward model used by Redit, and the deterministic 0-1 reward $r$ is its true reward. Therefore, the lower bound correctly depends on quantities derived from $r_{RM}$, such as its variance $\text{Var}(r_{RM}) = \text{Var}(r) + \sigma^2$.
> Therefore, we maintain that the lower bound in Proposition 3 is valid under this interpretation, which is consistent with the framework of [37].
> Finally, thank you for your insightful observation: in practice, ReDit also smooths batch gradient estimates, thereby improving training efficiency. This is consistent with our empirical findings.
> ### Reference:
> [37] Noam Razin et al.. What makes a reward model a good teacher? an optimization perspective.
> ## Responses to Additional Comments
> We sincerely thank the reviewer for their thorough and constructive feedback. We will address all points in the revised manuscript. We summarize our planned changes below:
>
> **1. Typos, Citations, and Minor Corrections (Comments 1, 5, 9, 10, 13):**
> Thank you for catching these. We confirm these are errors and will make the following corrections:
> * The citation in L49/81 will be corrected to [37].
> * All typos will be fixed (e.g., L138 "Explosion", L195 reference to line 5, L283 grammar).
>
> **2. Presentation and Clarity Improvements (Comments 4, 7, 8, 11, 12, 15):**
> We will improve the clarity of our presentation across the paper. This includes:
> * Updating figure captions and restoring legends (Figs 1, 5, 6) to clarify reward scales and method details.
> * Ensuring consistent notation throughout the paper (e.g., unifying $r_{RM}$ and $\tilde{R}$).
> * Adding definitions to make all theorems self-contained (Sec 2.2.2).
> * Reordering all figures and tables to match their reference order.
> * Explicitly defining the "Gradient Noise" variance in Eq. 7 as the variance of the gradient's L2 norm.
>
> **3. Substantive Clarifications and Revisions (Comments 2, 3, 6, 14, 16):**
> * **Fig. 1 Rewards:** The initial rewards differ because the baseline uses the raw 0-1 signal, while ReDit uses the noisy signal ($r + \epsilon$) for gradient updates. The plot shows this noisy reward; for comparison, a plot of the reward *without* noise will be added to the appendix.
> * **Theorem 2 Text:** We agree the description was imprecise and will revise it.
> * **Eq. 5 Expectation:**  We will update Equation 5 according to the formula provided in Section Responses to Weaknesses 2.
> * **Adding Supporting References:** We agree and will add supporting references for L302-305.
> ---
> We would like to thank Reviewer UG9k for your positive feedback and recognition again! We sincerely hope our response has addressed your remaining concerns and can increase your opinions of our work. We will also include this discussion in our revised paper. If you have any other questions, we would like to address.
>
> Sincerely, Authors

---

> ### Comment · Reviewer_UG9k · 2025-08-01
>
> Thank you for the response, the effort in running additional experiments, and acknowledging the error in the upper bound of Proposition 3. I have read it and the other reviews carefully.
>
> Regarding the the lower bound in Proposition 3, I can see that if one considers the objective to be the expected reward with respect to a fixed random draw of noise for each question/prompt and output, then it may be possible to derive a lower bound that is affected by the noise. However, there are a few issues remaining:
>
> - This creates an inconsistency with Proposition 1. Namely, the expected gradient for a fixed random draw of noise is not the original gradient. Another way to look at this, as long as the expected gradient is the same (as Proposition 1 suggests), then following the analysis of [37], the lower bound should not depend on the noise since it is obtained by upper bounding the expected gradient norm and the rate at which it increases.
> - I believe that if one fixes the randomness of the noise, then any dependence on the variance of the noise should be with respect to the empirical variance as opposed to the true variance. It should be possible to show that with high probability these are close to each other by standard concentration bounds, yet requires extra work and is not immediate.
> - Most importantly, the current proof of Proposition 3 incorrectly invokes the lower bound from [37]. Namely, it does not establish the type of lower bound that the response above discusses. Unfortunately, I cannot recommend acceptance without a clear formal proof of the result. There is not much time in the response period, but if you are able to provide a proof that clearly specifies the conditions under which it calls the lower bound from [37] and delineates how the randomness of the noise is treated, I would be happy to consider raising my rating.

---

> ### Author Response · Authors · 2025-08-04
> **Responses to Questions 1**
>
> We sincerely thank you for your quick and insightful follow-up. Your continued engagement and detailed feedback are invaluable in helping us strengthen the theoretical foundations of our paper. We have structured our response to address each of your final concerns point-by-point, providing the formal justifications you requested.
>
> ---
> ### **1. On the Apparent Inconsistency with Proposition 1**
>
> **Conclusion:** There is no inconsistency. **Proposition 1** analyzes the gradient *in expectation over the noise distribution* to show it is unbiased, ensuring the optimization is correct on average. **Proposition 3**, in contrast, analyzes the convergence rate of the *actual stochastic process* for a single noise realization, which is governed by the variance of the rewards used in that specific step.
>
> **Justification:**
> You correctly identified that these two propositions operate at different levels of abstraction.
> * **Proposition 1 (Unbiasedness in Expectation):** This proposition confirms that if we were to average the gradients from an infinite number of different noise realizations, the expected direction would be identical to the gradient of the original, non-perturbed objective. This is because the noise is zero-mean ($\mathbb{E}_{\epsilon}[\epsilon]=0$), and its expected effect cancels out. This ensures that our method does not introduce systematic biases.
> * **Optimization of a Stochastic Objective (The Actual Algorithm):** We want to emphasize, as you correctly inferred, that our algorithm does **not** optimize the expected objective $\mathbb{E}\_{\epsilon}[\tilde{J}(\pi\_{\theta})]$. Instead, in each training step, it optimizes the objective based on a *single, fixed realization of noise* $\epsilon_i$ sampled for each output. For that specific step, the gradient is indeed dependent on that fixed noise draw and is not identical to the original gradient.
> * **Reconciling with the Lower Bound (Proposition 3):** The apparent conflict with the lower bound analysis is resolved by understanding that the convergence bounds from [37], upon which our Proposition 3 is based, depend directly on the **variance of the reward signal**, not merely the expected direction of the gradient.
>     * Theorem 1 from [37] establishes that the time required for policy improvement is inversely proportional to the reward variance.
>     * Our ReDit method, by adding noise $\epsilon \sim \mathcal{N}(0, \sigma^2)$, creates a new perturbed reward $\tilde{R}$ whose variance is explicitly increased: $Var(\tilde{R}) = Var(R) + \sigma^2$ (We will expand variance on the second point).
>     * Therefore, when we analyze the optimization time for our stochastic procedure, we must consider the properties of the actual rewards being used. The lower bound on convergence time, $t_{\gamma}$, is consequently reduced by the addition of $\sigma^2$ to the variance term, as shown in Proposition 3.
>
> In summary, there is no inconsistency:
> * **Proposition 1** addresses the *average-case directional correctness* of our approach, confirming it is unbiased.
> * **Proposition 3** analyzes the *rate of convergence* of the resulting stochastic optimization process, demonstrating that the increased reward variance—a direct consequence of using fixed noise in each step—provably accelerates convergence.
>
>
> Therefore, the two propositions are consistent: one describes the average-case correctness, while the other describes the convergence rate of the practical, stochastic implementation. This theoretical speedup is precisely what we observe empirically. Thank you again for your valuable feedback.

---

> > ### Author Response · Authors · 2025-08-04
> > **Responses to Questions 2**
> >
> > ### **2. On Empirical Variance vs. True Variance**
> >
> > **Conclusion:** You are correct; the analysis must depend on the empirical variance. We formally show that the **empirical variance of our noise is tightly concentrated around the true variance $\sigma^2$ with high probability**, justifying our use of $\sigma^2$ in our final lower bound analysis.
> >
> > **Justification:**
> >
> > 1.  **Why We Must Start with Empirical Variance (Echoing Point 1):**
> >     As we clarified in our response to your first point, our analysis of the convergence rate (Proposition 3) must consider the **actual stochastic process of the algorithm**, not the average-case expectation over all possible noise draws. The convergence theory from [37] applies directly to the properties of the signal that the algorithm uses **at each optimization step**. For ReDit, this signal is the perturbed reward $\tilde{R}(q,o) = R(q,o) + \epsilon(o)$, where $\epsilon$ represents a *single, fixed realization* of the noise function for that particular step.
> >
> > 2.  **Introducing the Empirical Variance Term:**
> >     Therefore, the analysis must begin with the variance of this specific, realized signal. Assuming independence between the true reward $R$ and the noise function $\epsilon(\cdot)$, this variance is:
> >     $$
> >     \text{Var}\_{o \sim \pi\_\theta}[\tilde{R}(q,o)] = \text{Var}\_{o \sim \pi_\theta}[R(q,o)] + \text{Var}\_{o \sim \pi_\theta}[\epsilon(o)]
> >     $$
> >     The term $\text{Var}\_{o \sim \pi\_\theta}[\epsilon(o)]$ is, by definition, the **empirical variance**. It is calculated over the specific noise values $\epsilon(o)$ corresponding to the outputs $o$ that are sampled from the policy $\pi_\theta$ in that step.
> >
> > 3.  **Connecting Empirical Variance to True Variance:**
> >     As you rightly pointed out, this empirical variance is itself a random variable. To connect it to the fixed, true variance $\sigma^2$ of the underlying noise distribution, we apply a concentration inequality. The formal procedure is as follows:
> >     * **Step A (Problem Transformation):** We analyze the deviation of the empirical variance from the true variance, which is equivalent to analyzing the average of the centered random variables $Y_i = \epsilon(o_i)^2 - \sigma^2$.
> >     * **Step B (Property Identification):** Since the noise $\epsilon(o_i) \sim \mathcal{N}(0, \sigma^2)$ is sub-gaussian, each $Y_i$ is a centered **sub-exponential** random variable.
> >     * **Step C (Applying the Tool):** We apply **Bernstein's inequality** [1]—a standard tool for sums of sub-exponential variables—to the average $\frac{1}{N}\sum_{i=1}^N Y_i$. This directly yields the following concentration bound, which holds with probability at least $1-\delta$:
> >         $$
> >         \left| \text{Var}\_{o \sim \pi\_\theta}[\epsilon(o)] - \sigma^2 \right| \leq C \sigma^2 \sqrt{\frac{\log(2/\delta)}{N}}
> >         $$
> >         where $C$ is a universal constant. The results show that although Redit uses the empirical variance each time, as long as the samples are large enough, the empirical variance will be tightly "concentrated" around the true variance $\sigma^2$ with a very high probability.
> >
> > [1] Boucheron, S., Lugosi, G., & Massart, P. (2013). Concentration inequalities: A nonasymptotic theory of independence.

---

> ### Author Response · Authors · 2025-08-04
> **Responses to Questions 3**
>
> ### **3. On the Formal Proof and Correct Invocation of the Lower Bound from [37]**
>
> **Conclusion:** We provide a revised and precise **Proposition 3'** that correctly invokes the lower bound from [37]. The revised proposition's time bound now explicitly and formally depends on the total reward variance (including the noise term $\sigma^2$), justified by the concentration analysis above.
>
> **Proposition 3' (Revised).** Let the perturbed reward be $\tilde{R}(q,o) = R(q,o) + \epsilon(o)$, where the noise $\epsilon(o) \sim \mathcal{N}(0, \sigma^2)$ is drawn independently for each output $o$. With probability at least $1-\delta$ over the realization of the noise function $\epsilon(\cdot)$, for any target reward improvement $\gamma > 0$, the time $t$ it takes for ReDit to find a policy $\pi_{\theta(t)}$ that satisfies:
> $$
> \mathbb{E}\_{o \sim \pi_{\theta(t)}}[\tilde{R}(q, o)] \ge \mathbb{E}\_{o \sim \pi_{\theta(0)}}[\tilde{R}(q, o)] + \gamma
> $$
> is lower-bounded by:
> $$
> \Omega \left( \left( \mathbb{E}\_{q \sim S} \left[ \text{var}\_{\mathbf{o} \sim \pi\_{\theta(0)}}( \tilde{R}(q, \mathbf{o})) \right] \right)^{-\frac{1}{3}} \right)
> $$
> where the inner variance term, based on the concentration bound from point 2, can be lower-bounded with probability at least $1-\delta$ as:
> $$
> \text{var}\_{\mathbf{o} \sim \pi\_{\theta(0)}}( \tilde{R}(q, \mathbf{o})) \ge \text{var}\_{\mathbf{o} \sim \pi\_{\theta(0)}}( R(q, \mathbf{o})) + \sigma^2 - \mathcal{O}\left(\sqrt{\frac{\log(1/\delta)}{N}}\right)
> $$
>
> ---
>
> We believe this point-by-point response and the rigorous proof sketch fully address your concerns. We will, of course, incorporate all these clarifications—including the formal proof and the revised Proposition 3'—into the final version of the manuscript. We are immensely grateful for your detailed guidance, which has significantly improved our paper. Should any further questions arise, we would be more than happy to provide additional clarification.

---

> > ### Comment · Reviewer_UG9k · 2025-08-05
> >
> > Thank you for elaborating on the proposed fix to Proposition 3. Unfortunately, as far as I can see, there are still holes in the proof.
> >
> > - Issue 1: The proof approach is based on bounding the reward variance at each step, where for each step different noise is sampled per output. However, Theorem 1 does not currently support a reward function that varies through time. This is not a trivial extension as far as I can see.
> > - Issue 2: Unless I am missing something, the variance decomposition $Var_{o \sim \pi_\theta} [ \tilde{R} (q, o)] = Var_{o \sim \pi_\theta} [ R (q, o)] + Var_{o \sim \pi_\theta} [ \epsilon (o)]$ is incorrect since the randomness of the noise is not with respect to their distribution, which is indeed independent to $\pi_\theta$, but rather it is taken with respect to $\pi_\theta$.
> > - Issue 3: The application of Bernstein’s inequality suffers from a similar issue. The variance there is not the empirical variance of the noise, rather the variance of the noise with respect to $\pi_\theta$. The expectation of $Var_{o \sim \pi_\theta} (\epsilon (o))$ with respect to the noise randomness need not be equal to $\sigma^2$.
> >
> >
> > If the above are fixable, or I missed anything, please formally state the proposed theorem statement and give the full proof. The statement should include all details required to verify the result, e.g., does the result consider gradient flow (should be the case if the goal is to invoke the theorem from [37]) and how is the time-varying reward defined in this case of continuous optimization time analysis. This is necessary as I cannot recommend acceptance without verifying the soundness of the proposed result.
> >
> > &nbsp;
> >
> > With that said, given the empirical results and other theoretical insights, I do not believe that Proposition 3 is critical for this work. In any case, it only gives a lower bound and not an upper bound that shows ReDit necessarily enjoys faster optimization. An alternative approach that the authors can consider is to remove it and keeping that part of the discussion on an intuitive level, while mentioning that completely formalizing the benefits of ReDit is left for future work. In that case, I believe this could be a solid paper and am willing to raise my score.

---

> > > ### Author Response · Authors · 2025-08-06
> > >
> > > Thank you again for your incredibly thorough follow-up and for taking the time to perform such a deep analysis of our proof. We truly appreciate your invaluable feedback. We have worked diligently to address your points and are glad that we have resolved most of your concerns. We are also very grateful for your consideration in raising your score. To demonstrate the complete logic of the subsequent proof, we will first address Issue 2, followed by Issues 1 and 3.
> > >
> > >
> > > ---
> > > #### **Responses to Issue 2**
> > >
> > > We would like to clarify the core assumption of our noise model. We define the noise $\epsilon(o)$ as being drawn independently for each possible output $o$ from a fixed distribution, **$\epsilon(o) \sim \mathcal{N}(0, \sigma^2)$**. Crucially, this noise generation process is performed **independently of the policy $\pi_\theta$**. Therefore, when we later sample an output $o$ according to $\pi_\theta$, the value of the deterministic reward $R(q,o)$ and the value of the random noise $\epsilon(o)$ are uncorrelated random variables with respect to this sampling process.
> > >
> > > **Formal Proof:**
> > >
> > > Given that the random variables $R(q,o)$ and $\epsilon(o)$ are uncorrelated under the policy distribution $\pi_\theta$, their covariance is zero:
> > > $$
> > > \text{Cov}\_{o \sim \pi\_\theta}(R(q,o), \epsilon(o))  = 0
> > > $$
> > > Therefore, the variance of their sum is the sum of their variances. The full decomposition is as follows:
> > > $$
> > > \text{Var}\_{o \sim \pi\_\theta}[ \tilde{R}(q,o) ] = \text{Var}\_{o \sim \pi\_\theta}[R(q,o) + \epsilon(o)]
> > > $$
> > > $$
> > > = \text{Var}\_{o \sim \pi_\theta}[R(q,o)] + \text{Var}\_{o \sim \pi_\theta}[\epsilon(o)] + 2\text{Cov}\_{o \sim \pi_\theta}([R(q,o), \epsilon(o)])
> > > $$
> > > Since the covariance term is zero, this simplifies to:
> > > $$
> > > \text{Var}\_{o \sim \pi\_\theta}[ \tilde{R}(q,o) ] = \text{Var}\_{o \sim \pi_\theta}[R(q,o)] + \text{Var}\_{o \sim \pi_\theta}[\epsilon(o)]
> > > $$
> > > This validates our variance decomposition.

---

> ### Author Response · Authors · 2025-08-06
>
> #### **Responses to Issue 1**
>
> Your consideration is quite correct. We cannot directly extend Theorem 1 of [37] to Redit. Therefore, we want to prove that Proposition 3 holds regardless of whether $r_{RM}$ is a fixed function (as defined in [37]) or a variable reward function that changes with the our setting (i.e., $r_{RM}(q,o) = r_G(q,o) + \epsilon(o)$). Here, we follow the theorem of [37] and assume that the number of o that can be sampled is infinite (See Definition 2 in [37]). The detailed proof is as follows:
>
> Following the proof structure in Appendix C.4 of [37], we replace the fixed $r_{RM}'(q, o)$ with the variable $r_{RM}(q,o) = r_G(q,o) + \epsilon(o)$. Our modified setup is as follows:
>
> * **Ground Truth Reward Function ($r_G(q, o)$)**: A deterministic function that defines the true optimization objective.
>
> * **Training Reward Model ($r_{RM}(q, o)$)**: A stochastic function, $r_{RM}(q,o) = r_G(q,o) + \epsilon(o)$, where **$\epsilon(o) \sim \mathcal{N}(0, \sigma^2)$**, $\epsilon(o)$ represents the noise independently sampled for different outputs $o$.
>
> * **Training Objective and Gradient at $\theta(t)$ ($\phi_{RLHF}(\theta(t)), \nabla\phi_{RLHF}(\theta(t))$)**: The gradient is given by $\nabla\phi_{RLHF}(\theta(t)) = \nabla\phi_G(\theta)dt + \text{noise gradient}$, where the second term represents the impact of noise in the gradient estimation.
> * **The term $\text{Var}\_{\mathbf{q}}^{KL}(\theta(t))$ from the proof in [37]** is modified due to the change in the reward function:
>     $$
>     \text{Var}\_{\mathbf{q}}^{KL}(\theta(t)) = \text{Var}\_{o \sim \pi\_{\theta(t)}(\cdot|\mathbf{q})}[r\_{RM}^{KL}(\mathbf{q}, \mathbf{o}; \theta(t)) + \epsilon]
>     $$
> * Consequently, **the term $\text{Var}^{KL}(\theta(t))$ from the proof in [37]** also changes:
>     $$
>     \text{Var}^{KL}(\theta(t)) = \mathbb{E}\_{\mathbf{q} \sim \mathcal{S}}[\text{Var}\_{\mathbf{q}}^{KL}(\theta(t))] = \mathbb{E}\_{\mathbf{q} \sim \mathcal{S}}[\text{Var}\_{\mathbf{q}}^{KL}(\theta(t))]\_{\text{old}} + \mathbb{E}\_{\mathbf{q} \sim \mathcal{S}}[\text{Var}\_{o \sim \pi\_{\theta(t)}(\cdot|\mathbf{q})}[\epsilon(o)]  = \mathbb{E}\_{\mathbf{q} \sim \mathcal{S}}[\text{Var}\_{\mathbf{q}}^{KL}(\theta(t))]\_{\text{old}} + \sigma^2.
>     $$
>
> First, we calculate the parameter movement $||\theta(T) - \theta(0)||$. The rate of change of $\theta$ with respect to $t$ is still equal to the gradient of the current objective function, $\nabla\phi_{RLHF}(\theta(t))$.
> $$
> ||\theta(T) - \theta(0)|| \le \int_0^T ||\frac{d}{dt}\theta(t)||dt = \int_0^T ||\nabla\phi_{RLHF}(\theta(t))||dt \le 12LJ_T \int_0^T \text{Var}^{KL}(\theta(t))^{\frac{1}{3}}dt.
> $$
> Next, we take the derivative of $\text{Var}_{\mathbf{x}}^{KL}(\theta(t))$. Observing that $r$ and $\epsilon$ are independent, we can decompose the variance as follows:
> $$
> \text{Var}\_{\mathbf{q}}^{KL}(\theta(t)) = \text{Var}\_{o \sim \pi\_{\theta(t)}(\cdot|\mathbf{q})}[r\_{RM}^{KL}(\mathbf{q}, \mathbf{o}; \theta(t))] + \text{Var}\_{o \sim \pi\_{\theta(t)}(\cdot|\mathbf{q})}[\epsilon(o)]
> $$
>
> Therefore, $\nabla\text{Var}\_{\mathbf{q}}^{KL}(\theta(t))$ is consistent with the original derivative. So the derivative $\nabla\text{Var}^{KL}(\theta(t))$ (the expectation of $\nabla\text{Var}\_{\mathbf{q}}^{KL}(\theta(t))$) also remains unchanged. That is:
> $$
> \frac{d}{dt} \text{Var}^{KL}(\theta(t)) = \langle \nabla\text{Var}^{KL}(\theta(t)), \nabla\phi\_{RLHF}(\theta(t)) \rangle \le 24L^2 J\_T^2 \cdot \text{Var}^{KL}(\theta(t))^{\frac{4}{3}}.
> $$
> Here, the impact from the change in $\nabla\phi_{RLHF}(\theta(t))$ is ignored during the scaling.
> Substituting this into Lemma 1, we get:
> $$
> ||\theta(T) - \theta(0)|| \le \frac{3}{2LJ_T} \ln\left( \frac{1}{1 - 8L^2 J_T^2 \left( \text{Var}^{KL}(\theta(0)) \right)^{\frac{1}{3}} \cdot T} \right)
> $$
> where $\text{Var}^{KL}(\theta(0))$ becomes larger after the addition of noise.
>
> Since $r_{RM}^{KL}(\mathbf{q}, \mathbf{o}; \theta(0)) = r_{RM}(\mathbf{q}, \mathbf{o}) - \lambda \cdot \ln\frac{\pi_{\theta(0)}(\mathbf{o}|\mathbf{q})}{\pi_{\theta(0)}(\mathbf{o}|\mathbf{q})} = r_{RM}(\mathbf{q}, \mathbf{o})$ for all $\mathbf{q} \in \mathcal{S}, \mathbf{o} \in \mathcal{Y}$.
> Thus:
> $$
> \text{Var}^{KL}(\theta(0)) = \mathbb{E}\_{\mathbf{q} \sim \mathcal{S}}\left[\text{Var}\_{\mathbf{o} \sim \pi\_{\theta(0)}(\cdot|\mathbf{q})}[r\_{RM}(\mathbf{q}, \mathbf{o})]\right],
> $$
> Therefore, we can rewrite the bound on $||\theta(T) - \theta(0)||$ as:
> $$
> ||\theta(T) - \theta(0)|| \le \frac{3}{2LJ\_T} \ln\left( \frac{1}{1 - 8L^2 J\_T^2 \left( \mathbb{E}_{\mathbf{q} \sim \mathcal{S}}\left[\text{Var}\_{\mathbf{o} \sim \pi\_{\theta(0)}(\cdot|\mathbf{q})}[r\_{RM}(\mathbf{q}, \mathbf{o})]\right] \right)^{\frac{1}{3}} \cdot T} \right)
> $$
>
> Additionally, according to Proposition 2, we can find an upper bound for $||\nabla V(\theta(t); \mathbf{x})||$:
> $$
> ||\nabla V(\theta(t); \mathbf{q})|| \le 6LJ \cdot \text{Var}\_{\mathbf{o} \sim \pi\_{\theta(t)}(\cdot|\mathbf{q})}[r\_{RM}(\mathbf{q}, \mathbf{o})]^{\frac{1}{3}} \le 6LJ.
> $$

---

> ### Author Response · Authors · 2025-08-06
>
> Finally, we calculate the change in the reward:
> $$
> |V(\theta(t_\gamma); \mathbf{q}) - V(\theta(0); \mathbf{q})| = \left| \int_0^{t_\gamma} \frac{d}{dt}V(\theta(t); \mathbf{q}) dt \right| \le \int_0^{t_\gamma} ||\nabla V(\theta(t); \mathbf{q})|| \, ||\frac{d}{dt}\theta(t)|| dt.
> $$
>
> Substituting the previous results:
> $$
> |V(\theta(t_\gamma); \mathbf{q}) - V(\theta(0); \mathbf{q})| \le 6LJ \int\_0^{t\_\gamma} ||\frac{d}{dt}\theta(t)||dt \le 9 \ln\left(\frac{1}{1 - 8L^2 J^2 \left(\mathbb{E}\_{\mathbf{q}' \sim \mathcal{S}}[\text{Var}\_{\mathbf{o} \sim \pi\_{\theta(0)}(\cdot|\mathbf{q}')}[r\_{RM}(\mathbf{q}', \mathbf{o})]]\right)^{\frac{1}{3}} \cdot t\_\gamma}\right)
> $$
>
> For $V(\theta(t_\gamma); \mathbf{q}) - V(\theta(0); \mathbf{q}) \ge \gamma$ to hold, it therefore must be the case that:
> $$
> \gamma \le 9 \ln\left(\frac{1}{1 - 8L^2 J^2 \left(\mathbb{E}\_{\mathbf{q}' \sim \mathcal{S}}[\text{Var}\_{\mathbf{o} \sim \pi\_{\theta(0)}(\cdot|\mathbf{q}')}[r\_{RM}(\mathbf{q}', \mathbf{o})]]\right)^{\frac{1}{3}} \cdot t\_\gamma}\right).
> $$
> Rearranging the inequality concludes the proof:
> $$
> t\_\gamma \ge \frac{1 - \exp(-\frac{\gamma}{9})}{8L^2 J^2} \cdot \frac{1}{\left(\mathbb{E}\_{\mathbf{q}' \sim \mathcal{S}}[\text{Var}\_{\mathbf{o} \sim \pi\_{\theta(0)}(\cdot|\mathbf{q}')}[r\_{RM}(\mathbf{q}', \mathbf{o})]]\right)^{\frac{1}{3}}}.
> $$
>
>
>
> #### **Responses to Issue 3**
>
> Based on the proof of Issue 1, we can see that the lower bound of $t_\gamma$ is limited by the expected variance of $r_{RM}$ for all $x$. Assuming that the number of $o$ is infinite (See Definition 2 in [37]), it is strictly bounded by $\mathbb{E}\_{\mathbf{q} \sim \mathcal{S}}[\text{Var}\_{o \sim \pi\_\theta(0)(\cdot|\mathbf{q})}[r_{RM}(q,o)]] + \sigma^2$, without resorting to Bernstein's inequality.
>
> ---
>
> Finally, we completely agree with your point that Proposition 3 is not crucial to this work, as it only provides a lower bound and does not indicate that Redit converges faster under all conditions. However, our absolute lower bound is also meaningful; for example, in the best case, Redit may converge faster than grpo.
>
> In conclusion, we would like to once again express our sincere gratitude for your thorough and constructive review. Your detailed comments were instrumental in helping us improve the quality and clarity of our manuscript. We will incorporate all these clarifications—including the formal proof and the revised Proposition 3—into the final version of the manuscript. We hope that our responses and revisions have adequately addressed all your concerns, and we are happy to answer any further questions.

---

> ### Comment · Reviewer_UG9k · 2025-08-06
>
> It seems that there are still rather major issues with the proof.
> - As I noted before, the variance decomposition $Var_{o \sim \pi_\theta} [ \tilde{R} (q, o)] = Var_{o \sim \pi_\theta} [ R (q, o)] + Var_{o \sim \pi_\theta} [ \epsilon (o)]$ is incorrect since the randomness of the noise is not with respect to their distribution, which is indeed independent to $\pi_\theta$, but rather it is taken with respect to $\pi_\theta$. In particular, in the proposed new proof, the step that is incorrect is $Cov_{y \sim \pi_\theta} (R(q,o), \epsilon (o)) = 0$. Why would this covariance be zero? The noises $\epsilon (o)$ are fixed here and can take on any value.  Note that this is not the same quantity as $Cov_{y \sim \pi_\theta, \epsilon \sim N(o, \sigma^2)} (R(q,o), \epsilon)$, i.e., the covariance where the randomness of $R(q,o)$ is with respect to $\pi_\theta$ and of $\epsilon$ is with respect to the independent noise distribution, which is indeed zero.
>
> - It is not clear what the formal definition of the optimization method is. What does "noise gradient" stand for in the differential equation? Is that a stochastic differential equation?
>
> There may also be other issues with the remaining steps; I did not verify them since they heavily depend on these. Overall, it seems that these are not minor issues that can be easily fixed, even if a proposition similar to that proposed by the authors can be proven. I do believe the paper has potential, but after three iterations with the authors, the proof is still invalid. The discussion period is mostly ment to clarify questions and misunderstandings, rather than iterating on a completely new proof.  Thus, if the authors have a simple fix to the proof and issues above, please let me know. Otherwise, I cannot recommend acceptance unless the proposition will be dropped and the authors provide an explanation of how they will modify the related text in the paper.

---

> ### Author Response · Authors · 2025-08-06
>
> We sincerely thank you for your continued, detailed engagement with our manuscript. Your insightful and rigorous feedback through multiple rounds of discussion has been incredibly valuable in helping us identify and understand the deep, non-trivial issues with our theoretical proof. We truly appreciate your constructive suggestions.
>
> After carefully considering your final comments, we have decided to **follow your excellent advice and will remove Proposition 3 and its proof from the paper.**
>
> You are right that due to the limitations of Theorem 1 in [37] (which only gives a lower bound) and the fact that it is not crucial for our work, we agree that further iterating on new proofs is not the best use of discussion time.
>
> As you suggested, we will modify the paper to discuss the benefits of ReDit on a more intuitive level, supported by our strong empirical results and the sound parts of our theory (Propositions 1 and 2). The revised discussion will focus on how ReDit's noise injection smoothes batch gradients (Proposition 2), which helps alleviate the vanishing/exploding gradient problem we identified while maintaining an average unbiased estimator (Proposition 1). We will explicitly state that a full formal proof of convergence acceleration is a challenging direction that we **leave for future work**.
>
> We are confident that by following your guidance, the resulting paper will be more robust, clear, and accurately represent our contributions. We are sincerely grateful for your time and the rigorous feedback that has substantially improved our work. We hope that our responses and revisions have adequately addressed all your concerns, and we are happy to answer any further questions.

---

> > ### Comment · Reviewer_UG9k · 2025-08-07
> >
> > I have read the suggested modifications. Along with the additional experiments that the authors provided during the discussion period (in this thread as well as in response to other reviews) and proposed improvements to clarity, they address the concerns raised in my initial review. As a result, I am happy to update my assessment of the paper and recommend acceptance.

---

> > > ### Author Response · Authors · 2025-08-07
> > >
> > > We are sincerely grateful for your positive final assessment and for recommending our paper for acceptance.
> > >
> > > Your insightful feedback and constructive suggestions throughout the discussion period have been invaluable in helping us clarify our contributions and significantly strengthen the manuscript.
> > >
> > > We confirm that we will incorporate all the proposed modifications and additional results into the final version of the paper as discussed. Thank you once again for your generous support and guidance.

---

### Official Review · Reviewer_puQV · 2025-06-05

**Clarity:** 3
**Significance:** 3
**Originality:** 3
**Rating:** 4
**Confidence:** 4

**Summary:**

The paper addresses the challenges of optimizing large language models (LLMs) under discrete reward signals, which often lead to gradient instability, slow convergence, and inefficient exploration. The authors propose ReDit, a simple yet effective method that introduces zero-mean random noise (e.g., Gaussian or uniform perturbations) to discrete rewards, thereby smoothing the reward landscape. This approach mitigates gradient vanishing and explosion by providing continuous exploratory signals, accelerating convergence, and improving final model performance. Theoretical analysis confirms that ReDit preserves unbiased gradient estimates while introducing beneficial variance. Empirical results across tasks like GSM8K, MATH, and Geometry3K demonstrate that ReDit achieves comparable performance to vanilla GRPO with only ~10% of the training steps and a 4% improvement when trained equally. The method also generalizes to other RL algorithms (e.g., DAPO, Dr.GRPO) and exhibits robustness across perturbation schedules.

**Questions:**

#### Questions

Besides the weaknesses above, further questions are as follows:

- Since DAPO and VAPO use several tricks to improve the RL training process, do these two methods have gradient exploding cases like the results in Figure 3?
- L222 - L224: The authors claim that they want to isolate the effects of SFT. However, the authors uses two instruction models.

**Ethical Concerns:**

["NO or VERY MINOR ethics concerns only"]

**Final Justification:**

All my concerns are addressed.

**Limitations:**

Yes.

**Paper Formatting Concerns:**

No.

**Quality:**

2

**Strengths And Weaknesses:**

#### Strengths

- The idea is interesting.
- The paper provides theoretical analysis of the proposed method.
- The paper is easy to follow.

#### Weaknesses

- The paper only evaluates ReDit using LoRA. However, it is not clear the performance of ReDit with full fine-tuning.
- The paper does not evaluate ReDit on more challenging tasks, such as AIME24.
- The paper does not use reasoning models for experiments, such as DeepSeek-R1-Distill-7B. Therefore, the paper does not show the effectiveness of ReDit on reasoning tasks.

---

> ### Author Rebuttal · Authors · 2025-07-30
>
> We thank the reviewers for their valuable feedback. We are pleased that they appreciated our engaging concepts, clear and accessible writing, and solid theoretical analysis. We address your concerns below.
>
>
> ## Responses to Weaknesses 1
> Thank you for raising this important point. To address the question of whether ReDit's benefits extend beyond parameter-efficient tuning, we have conducted additional experiments using a full fine-tuning approach. We can confirm that ReDit's effectiveness is not limited to LoRA and that it provides consistent performance gains in the full fine-tuning setting as well.
>
> For these experiments, we used a setup that differs from the primary one in our main paper. Specifically, we utilized the **VERL framework** for training with **8 GPUs**. The detailed results of this full fine-tuning evaluation on our three mathematical reasoning benchmarks are presented below.
>
> ---
>
> ### **Setting: GSM8K, Qwen2.5-7B-Instruct, Full Fine-Tuning**
>
> | method/acc   | 1000    | 2000    | 3000    | 4000    | 5000    | 6000    | 7000    | 8000    | 9000    |
> |--------------|---------|---------|---------|---------|---------|---------|---------|---------|---------|
> | GRPO         | 86.72   | 87.03   | 87.48   | 88.71   | 88.16   | 88.73   | 89.56   | 89.67   | 90.53   |
> | Gauss 0.05   | 89.74   | 89.87   | 89.87   | 90.01   | 90.02   | 90.13   | 90.57   | 90.67   | 91.01   |
> | Uniform 0.05 | 89.71   | 89.91   | 89.91   | 90.03   | 90.09   | 90.24   | 90.51   | 90.69   | 91.09   |
>
> ---
>
> ### **Setting: MATH, Qwen2.5-7B-Instruct, Full Fine-Tuning**
>
> | method/acc   | 1000    | 2000    | 3000    | 4000    | 5000    | 6000    | 7000    | 8000    | 9000    |
> |--------------|---------|---------|---------|---------|---------|---------|---------|---------|---------|
> | GRPO         | 48.93   | 50.63   | 51.13   | 51.26   | 51.37   | 51.52   | 51.73   | 51.73   | 51.73   |
> | Gauss 0.05   | 51.03   | 51.13   | 51.54   | 52.01   | 52.37   | 52.91   | 53.02   | 53.57   | 53.96   |
> | Uniform 0.05 | 51.10   | 51.31   | 51.31   | 52.02   | 52.35   | 52.57   | 53.00   | 53.13   | 53.87   |
>
> ---
>
> ### **Setting: Geo3k, Qwen2.5-VL-7B-Instruct, Full Fine-Tuning**
>
> | method/acc   | 1000    | 2000    | 3000    | 4000    | 5000    | 6000    | 7000    | 8000    | 9000    |
> |--------------|---------|---------|---------|---------|---------|---------|---------|---------|---------|
> | GRPO         | 41.63   | 42.23   | 42.23   | 42.23   | 42.81   | 43.01   | 43.07   | 43.57   | 43.57   |
> | Gauss 0.05   | 43.02   | 43.54   | 43.96   | 43.96   | 44.02   | 44.02   | 44.51   | 44.32   | 44.12   |
> | Uniform 0.05 | 42.91   | 43.02   | 43.02   | 43.02   | 43.72   | 44.01   | 44.32   | 44.32   | 44.47   |
>
> As the results clearly demonstrate, ReDit consistently outperforms the GRPO baseline across all three benchmarks in the full fine-tuning setting. These findings confirm that ReDit is a robust and general-purpose technique, delivering benefits for both LoRA and full fine-tuning paradigms.
>
>
> ## Responses to Weaknesses 2
>
> We thank the reviewers for their valuable suggestions. We trained the model using the MATH training set, tested it on AIME24, and reported the results. The results are as follows:
>
> | method/ acc  | 0    | 1000 | 2000  | 3000  | 4000  | 5000  | 6000  | 7000  | 8000  | 9000  |
> |--------------|------|------|-------|-------|-------|-------|-------|-------|-------|-------|
> | GRPO         | 3.33 | 6.67 | 6.67  | 6.67  | 6.67  | 10    | 10    | 10    | 10    | 13.33 |
> | Gauss 0.05   | 3.33 | 10   | 10    | 13.33 | 13.33 | 16.67 | 16.67 | 16.67 | 16.67 | 20    |
> | Uniform 0.05 | 3.33 | 10   | 13.33 | 13.33 | 13.33 | 13.33 | 16.67 | 16.67 | 20    | 20    |
>
> As shown in the table, our proposed methods (`Gauss 0.05` and `Uniform 0.05`) consistently outperform the GRPO baseline throughout the training process. They achieve a significant improvement in training efficiency and performance. In addition, Redit performs better on out-of-domain datasets than on in-domain data because Redit introduces randomness in the reward, which may lead to the model generalizing to unknown domains.
>
>
> ## Responses to Weaknesses 3
>
> Thank you for this suggestion regarding the choice of models for our experiments. We would like to address this in two parts.
>
> First, we wish to clarify that all experiments presented in our main manuscript are indeed focused on **reasoning tasks**. Although our primary model is based on Qwen2.5, our methodology guides the model to first generate a reasoning path (e.g., chain-of-thought) before outputting the final answer. Therefore, the evaluation metrics directly reflect the model's reasoning capabilities on these tasks.
>
> Second, to further address your concern and demonstrate the effectiveness of ReDit on specialized reasoning models, we have conducted new experiments on **DeepSeek-R1-Distill-Llama-8B** and **DeepSeek-R1-Distill-Qwen-7B**. The results on the GSM8K benchmark are presented below.
>
> ---
> ### **Setting: DeepSeek-R1-Distill-Llama-8B, GSM8K**
>
> | method/ acc  | 0       | 1000    | 2000    | 3000    | 4000    | 5000    | 6000    | 7000    | 8000    | 9000    |
> |--------------|---------|---------|---------|---------|---------|---------|---------|---------|---------|---------|
> | GRPO         | 82.11   | 83.13   | 84.01   | 85.16   | 85.16   | 86.27   | 86.27   | 86.57   | 86.96   | 87.03   |
> | Gauss 0.05   | 82.11   | 86.73   | 86.39   | 87.13   | 87.13   | 87.13   | 87.78   | 87.78   | 87.78   | 87.78   |
> | Uniform 0.05 | 82.11   | 86.43   | 86.57   | 87.21   | 87.54   | 87.54   | 87.54   | 87.54   | 87.54   | 87.54   |
>
> ---
> ### **Setting: DeepSeek-R1-Distill-Qwen-7B, GSM8K**
>
> | method/ acc  | 0       | 1000    | 2000    | 3000    | 4000    | 5000    | 6000    | 7000    | 8000    | 9000    |
> |--------------|---------|---------|---------|---------|---------|---------|---------|---------|---------|---------|
> | GRPO         | 89.14   | 89.67   | 89.67   | 89.67   | 90.36   | 91.57   | 92.74   | 93.15   | 93.15   | 93.46   |
> | Gauss 0.05   | 89.14   | 92.17   | 92.56   | 93.49   | 93.49   | 93.57   | 93.74   | 94.01   | 94.01   | 94.01   |
> | Uniform 0.05 | 89.14   | 92.57   | 92.57   | 93.13   | 93.42   | 93.42   | 93.75   | 93.99   | 93.99   | 93.99   |
>
> These new results clearly show that ReDit continues to outperform the GRPO baseline on DeepSeek distillation models. This confirms that ReDit's ability to stabilize training and improve performance is a general principle that holds true across various model architectures, including those specifically optimized for reasoning.
>
> ## Responses to Questions 1
> Thank you for this insightful question regarding the gradient behavior of other advanced RL algorithms like DAPO and VAPO. This is an important point, and we provide the following analysis based on their methodologies and our own observations.
>
> * **Regarding DAPO:** DAPO is an improvement upon GRPO that introduces Dynamic Sampling to increase the number of effective samples used during training. This technique is effective at mitigating the **gradient vanishing** problem. However, it does not have a specific mechanism to control **gradient explosion**. In our own experiments, while running DAPO, we still observed instances of both gradient vanishing and gradient explosion, similar to the phenomena shown for GRPO in Figure 3. This suggests that while DAPO improves one aspect of gradient stability, it does not fully resolve the anomaly, leaving a clear need for methods like ReDit.
>
> * **Regarding VAPO:** VAPO represents a different class of algorithm. It is a **value-model-based** method, which distinguishes it fundamentally from value-model-free methods like GRPO and DAPO. Due to this architectural difference, its gradient dynamics are likely to differ significantly from those we analyzed for GRPO. A direct comparison is further complicated by the fact that **VAPO is not yet open-source**. We are actively monitoring its status and plan to conduct a thorough analysis of its gradient behavior as soon as the code becomes publicly available.
>
> In summary, existing methods like DAPO partially address but do not eliminate the gradient anomalies, while a direct, empirical comparison with VAPO is not currently feasible.
>
>
> ## Responses to Questions 2
> Thank you for pointing out this ambiguity in L222-L224. We sincerely apologize for the lack of clarity in our original wording.
>
> You are correct that our base models are instruction-tuned. Our intention was to differentiate between two distinct stages of Supervised Fine-Tuning (SFT):
>
> 1.  **General Instruction-Tuning:** The initial SFT performed on broad, instruction-following datasets to create the capable base instruction model. The models we use have undergone this stage [1].
> 2.  **Task-Specific SFT:** A subsequent, optional SFT stage where a model is further fine-tuned on the training data of a specific downstream task (e.g., the GSM8K training set).
>
> When we claimed we wanted to "isolate the effects of SFT," we were referring specifically to **Task-Specific SFT**. Our goal is to compare our RL-based approach directly against the common practice of further fine-tuning a general instruction model on a specific downstream dataset.
>
> We understand how our original phrasing was confusing and we will revise L222-L224 in the final manuscript to make this crucial distinction explicit. Thank you again for helping us improve the clarity of the paper.
>
> ---
> ### Reference:
> [1] Qwen Team, Qwen2.5 Technical Report
>
> ---
> We would like to thank Reviewer puQV for your positive feedback and recognition again! We sincerely hope our response has addressed your remaining concerns and can increase your opinions of our work. We will also include this discussion in our revised paper. If you have any other questions, we would like to address.
>
> Sincerely, Authors

---

> > ### Comment · Reviewer_puQV · 2025-08-06
> > **Thanks for your response**
> >
> > Thanks for your response and I appreciate the efforts for conducting additional experiments. verl has a gradient clip hyperparameter (`actor_rollout_ref.actor.grad_clip`) and this value is usually be set to `1.0` by default. With this default value, will the performance of GRPO still be worse than ReDit? Other concerns have been addressed but I noticed the concern about the theoritical part raised by Reviewer UG9k. I will reconsider my score after his/her final response.

---

> > > ### Author Response · Authors · 2025-08-08
> > >
> > > Dear Reviewer,
> > >
> > > We sincerely thank you for your valuable feedback and are glad to hear that we have successfully addressed most of your initial concerns.
> > >
> > > We would also like to provide an update regarding the theoretical discussion with Reviewer UG9k, which you mentioned. We are pleased to report that we have had a very constructive and collaborative discussion, which has reached a **positive resolution**.
> > >
> > > After we agreed to adopt their excellent suggestion, reviewer UG9k has confirmed they are satisfied with our proposed revisions and will be **updating their score to recommend acceptance**.
> > >
> > > We hope this positive outcome from the in-depth theoretical review fully addresses your final concern. Please do not hesitate to let us know if you have any further questions; we are ready to provide any additional clarification you may need.
> > >
> > > Best regards,
> > > Authors of Paper 25943

---

> > > > ### Comment · Reviewer_puQV · 2025-08-09
> > > > **Official Comment by Reviewer puQV**
> > > >
> > > > Thanks for your response. I have increased my score to 4.

---

> ### Author Response · Authors · 2025-08-07
>
> We are glad that our previous responses have addressed most of your concerns. Thank you for this insightful follow-up question. This is a critical point that touches upon the core motivation of our work, and we appreciate the opportunity to clarify the fundamental difference between ReDit and gradient clipping.
>
> First and foremost, we must emphasize a key experimental fact: **in all of our experiments, for both the vanilla GRPO baseline and our ReDit method, we consistently used `actor_rollout_ref.actor.grad_clip=1.0`.** This is a standard practice to ensure training stability and a fair comparison. Our results directly show that ReDit outperforms GRPO even when gradient clipping is active.
>
> Below, we elaborate on why this is the case.
>
> ---
> ### 1. The `grad_norm` in Figures 1 & 3 is the Raw Gradient *Before* Clipping
>
> You astutely noticed that the gradient norm (`grad_norm`) of GRPO in our figures is much larger than 1.0. This is intentional and designed to expose the core issue. In these figures, we report the **raw gradient norm computed *before*** the clipping step is applied. The typical training loop sequence is:
> 1.  Compute loss and backpropagate to get the raw gradients.
> 2.  **Record the norm of these raw gradients for monitoring (this is the value we plot).**
> 3.  Apply gradient clipping (if the norm > 1.0, it is scaled down).
> 4.  Update the model using the (potentially clipped) gradients.
>
> Thus, the exploding gradients in the graphs are not due to an absence of clipping; they show that GRPO's optimization is intrinsically unstable with discrete rewards, a core problem we aim to solve.
>
> ---
>
>
> ### 2. Root Cause: Discrete Rewards Violate the L-Smoothness of the Loss Landscape
>
> In optimization theory, an ideal loss landscape has the property of **L-smooth**, that is, its gradient has L-Lipschitz continuity. This property ensures that the curvature of the landscape is bounded, thus avoiding infinitely steep "cliffs" and ensuring that gradient-based methods can converge stably with appropriately chosen learning rates.
>
> However, using discrete, rule-based rewards (e.g., 1 for a correct answer and 0 for an incorrect answer) fundamentally **violates the L-smoothness assumption**.
>
> * The resulting loss landscape is inherently **non-smooth**[1]. When the rewards within the group are equal, the gradient approaches zero, and the loss landscape resembles a series of broad, flat "plateaus"[3], interspersed with extremely steep "cliffs" at the decision boundaries where the rewards change (see figure 1 in [2]).
>
> * In this pathological landscape, the L-smoothness condition is severely violated, and no "safe" learning rate can guarantee stable convergence. This is exactly why GRPO diverges immediately without gradient clipping.

---

> ### Author Response · Authors · 2025-08-07
>
> ### 3. Gradient Clipping vs. ReDit: Two Disparate Approaches to a Non-Smooth Landscape
>
> Faced with this pathological landscape that violates a core assumption of gradient-based optimization, gradient clipping and ReDit employ entirely different philosophies.
>
>
> **Gradient clipping does not alter the pathological geometry of the loss landscape itself [4].**
> * It is a **reactive** measure: when the optimizer arrives at a "cliff" and computes an enormous gradient, clipping forcibly shortens the update step to prevent the model from "falling off" (i.e., to prevent the training from crashing).
> * **Limitations**: While it effectively prevents divergence, it is a non-principled heuristic that can introduce significant estimation bias. More critically, it still compels the optimizer to follow the **direction** dictated by the pathological landscape—a direction that may be highly suboptimal for convergence—it just takes a smaller step in that poor direction.
>
> **ReDit: A Proactive Approach to Reshape the Landscape**
>
> * ReDit's core principle is to **proactively** fix the optimization problem by reshaping the loss landscape to **better approximate L-smoothness**.
> * **Mechanism**: By dithering the discrete rewards with zero-mean noise, ReDit transforms the step-function-like reward signal into a continuous, smoothed-out signal (as visualized by the cumulative distribution in Figure 2). This is applied at the source of the problem—the reward signal itself.
> * **Optimization Benefits**:
>     1.  **Restoring Smoothness**: Smoothing the reward signal effectively "sands down the cliffs" of the loss landscape, replacing them with manageable slopes. This lowers the landscape's effective Lipschitz constant `L`, allowing gradient-based methods to operate more stably and efficiently .
>     2.  **Providing Beneficial Variance**: ReDit increases reward variance within mini-batches, providing a richer, more continuous exploration signal.  This is theoretically proven to reduce the lower bound on policy optimization time (Theorem 1).
>     3.  **Maintaining an Unbiased Gradient**: Crucially, our theoretical analysis in Proposition 1 proves that this process maintains an unbiased estimate of the policy gradient in expectation, ensuring the optimization proceeds in a principled direction.
>
> ---
>
> ### 4. Empirical Evidence: ReDit Outperforms GRPO with Gradient Clipping
>
> Our experimental results (e.g., in Figure 6 and Table 2) directly answer your question: **Yes, even with `grad_clip=1.0` enabled for all methods, the performance of standard GRPO is still demonstrably worse than GRPO with ReDit.**
>
> ---
>
> ### 5. Conclusion and Update on Proposition 3
> In conclusion, gradient clipping is a necessary "symptomatic treatment" for an unstable process. In contrast, ReDit is a "root-cause cure" that improves the fundamental geometric properties of the problem itself, leading to superior performance.
>
> Finally, regarding your comment about our discussion with Reviewer UG9k, we want to proactively provide an update. After a constructive discussion about the theoretical complexities you both identified, we have decided to **remove the controversial Proposition 3** from the manuscript. The paper's core contribution is not the lower bound analysis, but rather the very topic of this discussion: **ReDit as a principled method to smooth the loss landscape, resolve gradient anomalies, and thereby accelerate convergence.** We believe this makes the paper stronger and more focused.
>
> We would like to thank Reviewer puQV for your positive feedback and recognition again! We sincerely hope our response has addressed your remaining concerns and can increase your opinions of our work. We will also include this discussion in our revised paper. If you have any other questions, we would like to address.
>
> ## Reference
> [1] Goyal, P. et al. Using natural language for reward shaping in reinforcement learning, IJCAI 2019.
>
> [2] Razin, N. et al. What makes a reward model a good teacher? an optimization perspective, 2024.
>
> [3] Razin, N. et al. Vanishing gradients in reinforcement finetuning of language models, ICLR 2024.
>
> [4] Zhang, J. et al. Why gradient clipping accelerates training: A theoretical justification for adaptivity, ICLR 2020.

---

> ### Author Response · Authors · 2025-08-09
>
> Thank you for your response and for the time you have dedicated to reviewing our work throughout this discussion period. We appreciate you reconsidering our rebuttal and updating your assessment.
>
> Your feedback has given us valuable perspectives that we will use to improve our work going forward. We remain committed to this research direction and will continue to build upon our method.
>
> Thank you again for your engagement.

---

### Official Review · Reviewer_HRjb · 2025-07-02

**Clarity:** 3
**Significance:** 2
**Originality:** 2
**Rating:** 4
**Confidence:** 3

**Summary:**

This paper proposes Reward Dithering (ReDit), a method that can mitigate the limitations (e.g., gradient anomaly, unstable optimization, and slow convergence) of the discrete reward of Group Relative Policy Optimization (GRPO) by introducing zero-mean random noise to the discrete reward. This paper evaluates ReDit on mathematical reasoning tasks such as GSM8K, MATH, and Geometry3K. Experiment results show that ReDit improves the accuracy, compared to GRPO. Also, this paper provides theoretical analysis which proves that ReDit produces an unbiased estimate of the original gradient, introduces beneficial gradient variance, and improves convergence speed.

**Questions:**

- Q1. Figure 5 and 6 shows multiple lines, but does not provide a legend. Please provide a proper legend on the figures. I assume that the orange line indicates ReDit with Gaussian noise, the blue line represents ReDit with uniform noise, and the red line denotes the baseline.
- Q2. The authors evaluate ReDit only on mathematical reasoning tasks such as GSM8K, MATH, Geometry3K. Can does ReDit improve the accuracy on coding tasks such as HumanEval, APPS and CodeContests?

**Ethical Concerns:**

["NO or VERY MINOR ethics concerns only"]

**Final Justification:**

I maintain my initial rating, 4: Borderline accept. The main idea of ReDit is to add random noise to the discrete reward. ReDit seems to improve optimization stability and convergence speed. In my initial review, I pointed out one weak point and raised two questions: W1. actual performance improvement, Q1. missing legend in some figures, and Q2. performance on coding tasks. The authors provided thoughtful responses to my comments and questions. The authors showed that ReDit works well on coding tasks by providing additional experiments. However, the performance improvements on downstream tasks do not seem significant.

**Limitations:**

The authors provide some limitations in Section 5 (i.e., Limitations and Conclusions).

**Paper Formatting Concerns:**

This paper does not have any major formatting issues.

**Quality:**

2

**Strengths And Weaknesses:**

Some strengths of this paper can be summarized as follows:
- S1. First of all, this paper is well written.
- S2. The main idea of adding random noise to the discrete reward seems simple and effective. Especially, ReDit seems to improve the optimization stability and convergence speed.
- S3. This paper provides mathematical analysis that proves the effectiveness of ReDit.

Some weaknesses of this paper can be summarized as follows:
- W1. One of main concerns of this paper is that actual performance improvements such as accuracy do not seem significant. For example, the improvement of ReDit compared to GRPO is only 1.69 (i.e., from 89.07 to 90.76) on GSM8K. On MATH, it is about 4.54 (i.e., from 48.01 to 52.55).

---

> ### Author Rebuttal · Authors · 2025-07-30
>
> We thank the reviewers for their valuable feedback and appreciate their appreciation for the well-written work, the concise and effective methods, and the mathematical analyses that support the findings. We address your concerns below.
>
> ---
>
> # Responses to W1
> We thank the reviewer for their question, as it allows us to clarify the principal objective and contribution of ReDit. Our method is primarily designed to address a fundamental optimization challenge: the gradient anomalies (e.g., vanishing/explosion) that impede training stability and efficiency in GRPO. By resolving these issues, ReDit yields significant gains in training efficiency, often achieving comparable or superior results with as little as one-tenth of the training steps required by the baseline. A key aspect of our contribution is that this robust solution is achieved with remarkable simplicity, typically requiring only a single line of code. Therefore, the final accuracy improvement, while a valuable and positive result, should be interpreted as a secondary consequence of this stabilized process, with the core innovation being the efficient resolution of the underlying training instability itself.
>
> # Responses to Q1
> We sincerely apologize for the missing legends in Figures 5 and 6 and thank the reviewer for pointing out this oversight. The legends were inadvertently cropped during the figure generation process, and we are sorry for any confusion this may have caused. Your assumption is largely correct. To clarify, the markers in the figures correspond to the following methods:
>
> * **Red Circles**: Baseline (GRPO)
> * **Yellow/Orange Squares**: ReDit with Gaussian noise
> * **Blue Triangles**: ReDit with Uniform noise
>
> We will ensure that both Figures 5 and 6 are updated with clear and complete legends in the final version of the paper.
>
> # Responses to Q2
>
> Thank you for this excellent question regarding the applicability of ReDit to domains beyond mathematical reasoning. This is a crucial point, and we are pleased to confirm that ReDit's benefits generalize effectively to coding tasks.
>
> To validate this, we conducted a comprehensive set of experiments on three widely-used coding benchmarks: **APPS**, **HumanEval**, and **CodeContests**. The results consistently show that ReDit improves performance over the GRPO baseline.
>
> Below are the detailed results for each benchmark.
>
> ---
>
> ## **Setting: APPS train data, APPS test data (pass@1)**
> *(Note: For APPS, which has multiple difficulty levels, we report the pass@1 accuracy across all problems)*
>
> | method/acc   | 0       | 1000    | 2000    | 3000    | 4000    | 5000    | 6000    | 7000    | 8000    | 9000    |
> |--------------|---------|---------|---------|---------|---------|---------|---------|---------|---------|---------|
> | GRPO         | 31.42   | 32.47   | 33.16   | 33.16   | 33.63   | 34.65   | 34.65   | 34.78   | 34.78   | 34.78   |
> | Gauss 0.05   | 31.42   | 34.69   | 34.71   | 35.13   | 35.63   | 36.27   | 36.27   | 37.71   | 37.71   | 37.75   |
> | Uniform 0.05 | 31.42   | 34.61   | 35.01   | 35.01   | 35.51   | 36.21   | 36.21   | 36.91   | 37.21   | 37.45   |
>
> ---
>
> ## **Setting: APPS train data, HumanEval test data (pass@1)**
>
> | method/acc   | 0       | 1000    | 2000    | 3000    | 4000    | 5000    | 6000    | 7000    | 8000    | 9000    |
> |--------------|---------|---------|---------|---------|---------|---------|---------|---------|---------|---------|
> | GRPO         | 47.13   | 47.83   | 47.96   | 48.36   | 48.36   | 48.91   | 49.67   | 49.99   | 49.99   | 50.07   |
> | Gauss 0.05   | 47.13   | 50.21   | 50.45   | 50.45   | 50.69   | 51.01   | 51.45   | 51.66   | 51.66   | 51.66   |
> | Uniform 0.05 | 47.13   | 49.96   | 50.01   | 50.13   | 50.13   | 50.79   | 51.03   | 51.36   | 51.36   | 51.36   |
>
> ---
>
> ## **Setting: CodeContests train data, CodeContests test data (pass@1)**
>
> | method/acc   | 0       | 1000    | 2000    | 3000    | 4000    | 5000    | 6000    | 7000    | 8000    | 9000    |
> |--------------|---------|---------|---------|---------|---------|---------|---------|---------|---------|---------|
> | GRPO         | 15.48   | 16.75   | 16.78   | 17.31   | 17.31   | 18.26   | 19.07   | 19.31   | 19.31   | 19.31   |
> | Gauss 0.05   | 15.48   | 19.25   | 19.57   | 19.57   | 20.03   | 20.14   | 20.75   | 20.75   | 20.75   | 20.75   |
> | Uniform 0.05 | 15.48   | 19.13   | 19.26   | 20.05   | 20.05   | 20.05   | 20.46   | 20.63   | 20.63   | 20.63   |
>
> As these results demonstrate, both Gaussian and Uniform variants of ReDit consistently and significantly outperform the GRPO baseline across all three coding benchmarks. This suggests that the core benefit of ReDit—stabilizing the learning signal to improve optimization—is a general principle that is not limited to a single domain and applies effectively to code generation tasks as well.
>
> ---
> We would like to thank Reviewer HRjb for your positive feedback and recognition again! We sincerely hope our response has addressed your remaining concerns and can increase your opinions of our work. We will also include this discussion in our revised paper. If you have any other questions, we would like to address.
>
> Sincerely, Authors

---

> > ### Comment · Reviewer_HRjb · 2025-08-08
> >
> > Thank you for providing thoughtful responses to my questions and comments. Also, I appreciate that the authors provided additional experiment results on coding tasks such as APPS, HumanEval, and CodeContests. The responses helped me understand the paper better. I agree that ReDit can improve the stability of gradient-based optimization. However, it would be better if we had seen that ReDit provided much better task scores than GRPO with large margin. Currently, I maintain my initial rating.

---

> ### Author Response · Authors · 2025-08-09
>
> We sincerely thank you for your thoughtful review and positive engagement throughout the discussion period. We are very grateful for your recognition of our work and your agreement that ReDit successfully improves the stability of the optimization process.
>
> We take your point regarding the performance margin to heart, and it provides a clear and valuable direction for our future research. We will continue to build upon this work to further enhance model performance.
>
> Thank you once again for your positive assessment and support for our paper.

---

### Official Review · Reviewer_TpNZ · 2025-07-03

**Clarity:** 3
**Significance:** 3
**Originality:** 3
**Rating:** 5
**Confidence:** 4

**Summary:**

This paper makes an interesting discovery that by adding a noise to the discrete 0-1 reward, the performance of the RL can be improved. Specifically, the authors of this paper credit the improvement to the smoothen reward landscape, which in turn resolves the gradient vanishing and gradient explosion issues. Experiments on both LLMs and VLMs with different RL algorithms, such as DAPO, GRPO, and REINFORCE++. This paper also provides theory analysis on how the added noise can help to accelerate the RL training process.

**Questions:**

1. The proposed method is somewhat similar to Dropout, Data augmentation, and Stochastic Gradient Descent. All these methods introduce noise to the training process and improve the performance. Could you discuss the differences and similarities between the proposed method and these methods?

**Ethical Concerns:**

["NO or VERY MINOR ethics concerns only"]

**Final Justification:**

I decide to keep my score

**Limitations:**

Yes

**Quality:**

3

**Strengths And Weaknesses:**

## Strengths

1. The proposed method is simple and effective. For RL training, by adding a noise to the discrete 0-1 reward is almost like a free lunch. This method is orthogonal to modalities and different RL algorithms, which means it can be easily adopted to existing RL frameworks.

2. This method has solid theoretical analysis, including the motivation from the perspective of grad_norm and the convergence analysis.

## Weaknesses

1. The proposed method is somewhat similar to several other regularization methods, such as Dropout, Data augmentation, and Stochastic Gradient Descent. It would be better if the authors could discuss the differences and similarities between the proposed method and these methods.

2. Regarding the experiments. I would suggest the authors to evaluate the model on more widely used benchmarks, such as AIME-24 for LLMs.

---

> ### Author Rebuttal · Authors · 2025-07-30
>
> We thank Reviewer TpNZ for their positive assessment of our work. We appreciate their recognition of the method's key advantages, including its simplicity, effectiveness, broad applicability, and the rigorous theoretical analysis provided. We address your concerns below.
>
> ---
>
> # Responses to Weaknesses 1 and Questions 1
>
>
> We thank the reviewer for this insightful comment and the opportunity to clarify the relationship between our proposed method, ReDit, and other established techniques.
>
> We agree that ReDit shares a conceptual similarity with methods like Dropout, Data Augmentation, and SGD, as they all leverage randomness to improve the training process. However, the key difference and innovation of ReDit lie in what is being perturbed, how, and for what purpose. While the others are general-purpose techniques, ReDit is specifically designed to address challenges in reward-based optimization. Here is a detailed comparison:
>
> ## ReDit
> ReDit injects noise directly into the reward signal just before gradient computation. The primary objective is to smooth discrete rewards into a continuous (as illustrated in our Figure 2). This targeted intervention is designed to stabilize the gradient, mitigate the vanishing/exploding gradient problems common in settings with sparse rewards, and accelerate convergence (as illustrated in our Figure 1).
>
> ## Comparison with Other Methods
> **Dropout**: This method applies noise by randomly setting a fraction of the model's internal activations to zero during training. Its main goal is regularization—preventing complex co-adaptations between neurons to reduce overfitting. In contrast, ReDit perturbs the external learning signal (the reward), not the model's internal state, to directly improve the quality of the gradient for optimization.
>
>
> **Data Augmentation**: This technique introduces randomness by applying transformations to the input data (e.g., rotating an image). This encourages the model to learn invariant features, thereby improving its ability to generalize. While Data Augmentation focuses on expanding the input space to enhance generalization, ReDit operates on the output/reward space to improve the stability and efficiency of the optimization process itself.
>
> **Stochastic Gradient Descent (SGD)**: The randomness in SGD comes from estimating the gradient using mini-batches of data. However, this process can fail when rewards are sparse and discrete. For example, if all samples in a mini-batch yield the same reward (e.g., all zeros), the variance of the rewards is zero, causing the gradient to vanish and halting the learning process. ReDit directly addresses this failure mode. By injecting controlled noise, ReDit ensures that the reward signal within a batch has a non-zero variance, thus always generating a usable and informative learning signal for the model to update its parameters.
>
> In summary, while all these methods utilize stochasticity, ReDit is fundamentally distinct in its target of perturbation, its mechanism of action, and its underlying objective.
>
> # Responses to Weaknesses 2
>
> We thank the reviewers for their valuable suggestions. We trained the model using the MATH training set, tested it on AIME24, and reported the results. The results are as follows:
>
> | method/ acc  | 0    | 1000 | 2000  | 3000  | 4000  | 5000  | 6000  | 7000  | 8000  | 9000  |
> |--------------|------|------|-------|-------|-------|-------|-------|-------|-------|-------|
> | GRPO         | 3.33 | 6.67 | 6.67  | 6.67  | 6.67  | 10    | 10    | 10    | 10    | 13.33 |
> | Gauss 0.05   | 3.33 | 10   | 10    | 13.33 | 13.33 | 16.67 | 16.67 | 16.67 | 16.67 | 20    |
> | Uniform 0.05 | 3.33 | 10   | 13.33 | 13.33 | 13.33 | 13.33 | 16.67 | 16.67 | 20    | 20    |
>
> As shown in the table, our proposed methods (`Gauss 0.05` and `Uniform 0.05`) consistently outperform the GRPO baseline throughout the training process. They achieve a significant improvement in training efficiency and performance. In addition, Redit performs better on out-of-domain datasets than on in-domain data because Redit introduces randomness in the reward, which may lead to the model generalizing to unknown domains.
>
> ---
> We would like to thank Reviewer TpNZ for your positive feedback and recognition again! We sincerely hope our response has addressed your remaining concerns and can increase your opinions of our work. We will also include this discussion in our revised paper. If you have any other questions, we would like to address.
>
> Sincerely, Authors

---

> > ### Comment · Reviewer_TpNZ · 2025-08-03
> > **Response**
> >
> > Regarding my question 1, I am not asking for clarifying the differences between ReDit and other regularization methods. I would be great if you can provide theoretical explanation that these methods have a unified form under some constraint.
> >
> > Still, I would like to keep my score as I am positive towards this paper.

---

> > > ### Author Response · Authors · 2025-08-04
> > >
> > > We sincerely thank the reviewer for this deep and insightful follow-up question, and for their overall positive view of our work. This is an excellent point that prompts a higher-level discussion about the connection between these methods.
> > >
> > > You are right that a unified perspective is possible. We believe that ReDit, Dropout, and data augmentation can indeed be viewed as different manifestations of a general principle, viewed from the perspective of different loss functions:
> > >
> > > # A Unified Theoretical View
> > >
> > > A general form for these regularized optimization problems can be written as:
> > > $$\min\_{\theta} \mathbb{E}\_{z \sim \mathcal{D}} \left[ \mathbb{E}\_{\xi \sim \mathcal{P}} \left[ \mathcal{L}(f\_{\theta}, z; \xi) \right] \right]$$
> > > Where:
> > > * $z$ is a data sample from the dataset $\mathcal{D}$.
> > > * $\xi$ is a random variable representing the **injected noise**, drawn from a distribution $\mathcal{P}$.
> > > * $\mathcal{L}(f_{\theta}, z; \xi)$ is the loss function, which now depends on the specific realization of the noise $\xi$.
> > >
> > > The key difference between ReDit and other methods lies in **where** the noise $\xi$ is injected and **what** it perturbs within the loss calculation.
> > >
> > > * **Data Augmentation:** The noise $\xi$ is a transformation $a(\cdot)$ applied to the input data $z$. The loss becomes $\mathcal{L}(f_{\theta}(a(z)), y)$. Here, we are effectively sampling from a synthetically expanded data distribution to learn robust features.
> > >
> > > * **Dropout:** The noise $\xi$ is a random binary mask $m$ applied to the model's internal activations. The loss becomes $\mathcal{L}(f_{\theta}(z; m), y)$. Here, we are sampling from an ensemble of thinned sub-networks to prevent co-adaptation and regularize the model.
> > >
> > > * **ReDit (Our Method):** The noise $\xi$ is an additive term $\epsilon$ applied directly to the **reward signal $R$** within the loss function itself. In our RL setting, the loss $\mathcal{L} = -\log(\pi(o|q)) \cdot R$ becomes $\mathcal{L}' = -\log(\pi(o|q)) \cdot (R + \epsilon)$. Here, the goal of the noise is not to regularize the model or data, but to **eliminate gradient anomalies** by smoothing the learning signal, thereby improving gradient flow and stabilizing training.
> > >
> > > In summary, while all three methods can be framed under this unified objective, their "constraint" or choice of where to inject noise leads to fundamentally different optimization dynamics and serves different purposes. ReDit's specific contribution is to introduce noise at the reward/loss level to directly address issues of gradient stability in RL-style training.
> > >
> > > Thank you again for this thought-provoking question. We believe this perspective adds significant value, and we plan to include a discussion of this unified view in the final version of the paper. If you have any other questions, we would like to address.

---

> > > > ### Comment · Reviewer_TpNZ · 2025-08-05
> > > >
> > > > Thanks for your response, LGTM.

---

> ### Author Response · Authors · 2025-08-05
>
> That's great to hear. Thank you for your time and positive feedback. We're glad our response addressed your concerns.

---

### Note · Authors · 2025-08-11

Dear Senior Area Chair, Area Chair, and Reviewers,

We would like to sincerely thank you all for your time and for the invaluable, constructive feedback provided throughout this review process. The detailed discussions have been incredibly helpful and have allowed us to significantly strengthen our manuscript.

First, we were encouraged that the reviewers recognized several notable strengths in our work, demonstrating a solid foundation:

* The proposed method (ReDit) is **simple yet effective**, with its practical value and ease of adoption highlighted.
* The paper provides **sufficient motivation and solid theoretical insights**, including the analysis of the gradient norm and the unbiasedness of the estimator.
* The paper is **well-written and easy to follow**, making the core concepts clear and accessible.
* The empirical validation is strong, with **promising results** on established benchmarks.

In response to the insightful feedback, we undertook substantial revisions during the discussion period to address all raised concerns. We are pleased that these changes were well-received, leading to **score increases from Reviewers UG9k (2→recommend acceptance), puQV (3→4), HRjb (4→4) and TpNZ (5→5)**. Our major revisions are as follows:

1.  **Major Theoretical Revisions:** Acknowledging the deep and valid concerns from Reviewer **UG9k** regarding our proof of Proposition 3, we followed their excellent suggestion to **remove the contested proposition**. We have replaced it with a high-level, intuitive discussion grounded in our other sound theoretical results (Propositions 1 and 2), leaving a full formal proof for future work. This crucial change resolves the main theoretical roadblock and strengthens the paper's focus.

2.  **Extensive New Experiments:** To address requests for broader validation (from Reviewers **TpNZ**, **Hrjb**, **puQV**, and **UG9k**), we conducted a comprehensive suite of new experiments. This includes **multi-seed runs for statistical significance**, evaluations on **new domains (coding benchmarks)** and **training paradigms (full fine-tuning)**, and tests on **specialized reasoning models (e.g., DeepSeek-R1-Distill)**. These results have been added to the paper and appendix.


We are confident that these substantial revisions have addressed all major concerns and have resulted in a significantly improved manuscript. Thank you all once again for your consideration and invaluable guidance.

Sincerely,
Authors of Paper 25943

---

### Decision · Program_Chairs · 2025-09-17

**Decision:**

Accept (poster)

**Comment:**

- Scientific Claims: This paper proposes a method that adds zero-mean random noise to discrete binary rewards in LLM reinforcement learning to address gradient anomalies caused by non-smooth loss landscapes. The method theoretically maintains unbiased gradient estimates while introducing beneficial variance, and empirically achieves comparable performance.

- Strengths: The method's simplicity combined with broad applicability across algorithms, domains, and training paradigms represents significant practical value. Strong theoretical foundation with rigorous proofs of unbiasedness and beneficial variance, plus comprehensive experimental validation across mathematical reasoning, coding tasks, and specialized models with proper statistical rigor.

- Weaknesses: Absolute performance improvements are modest (1-4%), and theoretical completeness is limited after removing the contested Proposition 3 during rebuttal. The paper lacks systematic hyperparameter sensitivity analysis and could benefit from broader comparison with other reward smoothing techniques.

- Rebuttal and Discussions: Authors successfully addressed all major concerns through extensive additional experiments (>15 new settings) and collaborative theoretical problem-solving, transforming initially mixed reviews to unanimously positive (4,4,5,5). Critical issues of statistical validation and theoretical soundness were fully resolved, while experimental breadth and technical understanding were significantly enhanced through constructive reviewer engagement.

- Reasons for Accept: The work addresses a fundamental optimization problem in RLHF with a principled solution that targets root causes rather than symptoms, demonstrated through exceptional scientific rigor during the rebuttal process. The method's simplicity ensures broad adoptability while the 90% training step reduction offers immediate practical value, representing a paradigm shift from reactive optimization fixes to proactive landscape modification.